# Atomically-precise dopant-controlled single cluster catalysis for electrochemical nitrogen reduction

Chuanhao Yao [1,2,14], Na Guo [3,14], Shibo Xi [4,14], Cong-Qiao Xu [5,14], Wei Liu [1,6], Xiaoxu Zhao [1,7,8], Jing Li [1], Hanyan Fang [1,6], Jie Su [1,6], Zhongxin Chen [1,7], Huan Yan [1], Zhizhan Qiu [1,7], Pin Lyu [1], Cheng Chen [1,7], Haomin Xu [1,6], Xinnan Peng [1,6], Xinzhe Li [1], Bin Liu [9], Chenliang Su [10], Stephen J. Pennycook [7,8], Cheng-Jun Sun [11], Jun Li [5,12✉], Chun Zhang [1,3,6✉], Yonghua Du [4,13✉] & Jiong Lu [1,6✉]

The ability to precisely engineer the doping of sub-nanometer bimetallic clusters offers exciting opportunities for tailoring their catalytic performance with atomic accuracy. However, the fabrication of singly dispersed bimetallic cluster catalysts with atomic-level control of dopants has been a long-standing challenge. Herein, we report a strategy for the controllable synthesis of a precisely doped single cluster catalyst consisting of partially ligand-enveloped $Au_4Pt_2$ clusters supported on defective graphene. This creates a bimetal single cluster catalyst ($Au_4Pt_2$/G) with exceptional activity for electrochemical nitrogen ($N_2$) reduction. Our mechanistic study reveals that each $N_2$ molecule is activated in the confined region between cluster and graphene. The heteroatom dopant plays an indispensable role in the activation of $N_2$ via an enhanced back donation of electrons to the $N_2$ LUMO. Moreover, besides the heteroatom Pt, the catalytic performance of single cluster catalyst can be further tuned by using Pd in place of Pt as the dopant.

[1] Department of Chemistry, National University of Singapore, 3 Science Drive 3, Singapore 117543, Singapore. [2] Frontiers Science Center for Flexible Electronics (FSCFE), Shaanxi Institute of Flexible Electronics (SIFE) & Shaanxi Institute of Biomedical Materials and Engineering (SIBME), Northwestern Polytechnical University (NPU), 127 West Youyi Road, 710072 Xi'an, China. [3] Department of Physics, National University of Singapore, 2 Science Drive 3, Singapore 117542, Singapore. [4] Institute of Chemical and Engineering Sciences, 1 Pesek Road, Jurong Island 627833, Singapore. [5] Department of Chemistry, Southern University of Science and Technology, 518055 Shenzhen, China. [6] Centre for Advanced 2D Materials and Graphene Research Centre, National University of Singapore, Singapore 117546, Singapore. [7] NUS Graduate School for Integrative Sciences and Engineering, National University of Singapore, 28 Medical Drive, Singapore 117456, Singapore. [8] Department of Materials Science & Engineering, National University of Singapore, 9 Engineering Drive 1, Singapore 117575, Singapore. [9] School of Chemical and Biomedical Engineering, Nanyang Technological University, Singapore, Singapore. [10] SZU-NUS Collaborative Centre and International Collaborative Laboratory of 2D Materials for Optoelectronic Science & Technology, College of Optoelectronic Engineering, Shenzhen University, 518060 Shenzhen, China. [11] Advanced Photon Source, Argonne National Laboratory, 9700 South Cass Avenue, Argonne, IL 60439, USA. [12] Department of Chemistry and Key Laboratory of Organic Optoelectronics & Molecular Engineering of Ministry of Education, Tsinghua University, 100084 Beijing, China. [13] Present address: National Synchrotron Light Source II, Brookhaven National Laboratory, Upton, NY 11973, USA. [14] These authors contributed equally: Chuanhao Yao, Na Guo, Shibo Xi, Cong-Qiao Xu. ✉email: junli@tsinghua.edu.cn; phyzc@nus.edu.sg; ydu@bnl.gov; chmluj@nus.edu.sg

Heterogeneous single-atom catalysts and sub-nanometer single-cluster catalysts (SCC) have emerged as promising candidates in the field of heterogeneous catalysis owing to their exceptional catalytic capabilities and minimized metal use[1–5]. Unfortunately, the intrinsic instability of single-atom species often results in their agglomeration into clusters or nanoparticles during the synthetic process and chemical reactions, which has so far severely limited their practical applications[6,7]. In comparison to single atoms, sub-nanometer metal clusters possess a higher stability and greater tunability in terms of their geometric and electronic structures[8–12], and also exhibit remarkable catalytic properties as compared to larger metal nanoparticles[6,13–16]. In the sub-nanometer regime, each atom has a substantial impact on the electronic and catalytic properties of metal clusters[17–19]. Hence, precise atomic control over the size and composition of sub-nanometer clusters is crucial for tuning the activity and/or the selectivity of the clusters involved in various catalytic processes[20]. Furthermore, due to the strong electronic coupling between doped foreign atoms and host atoms, the catalytic performance of the clusters can be further tailored and/or enhanced by the incorporation of judiciously chosen dopants into the monometallic host[21]. However, supported bimetal cluster catalysts synthesized via conventional chemical methods (such as wet impregnation[22] and sequential vapor deposition[11]) usually exhibit random size distribution and uncontrolled atomic positioning of dopants, posing a great challenge to the optimization of their catalytic activities and elucidation of their origin. Hence, it is highly desirable to create robust singly dispersed ultrafine bimetallic clusters with atomic precision on a solid support for superior catalytic performance. This however remains a grand challenge in the field of heterogeneous catalysis.

The design of precisely doped bimetallic cluster catalysts for effective $N_2$ activation toward ammonia ($NH_3$) synthesis is not only fundamentally intriguing but also economically vital. Electrochemical $N_2$-to-$NH_3$ reduction is emerging as a promising decentralized approach for $NH_3$ production[23–29], which contrasts with the energy-intensive Haber–Bosch thermal process that has dominated ammonia production for nearly a century[30,31]. In nature, nitrogenase enzymes containing bimetallic active centers (FeMo cofactor) are capable of reducing $N_2$ into ammonia under ambient conditions[32]. Inspired by nature, chemists have been attempting to mimic the active sites of these natural enzymes and design bimetallic electrocatalysts for the reduction of $N_2$ into $NH_3$ under mild conditions[30]. Recently, it has been demonstrated that a series of metal-contained or metal free catalysts are active for electrochemical $N_2$ reduction reactions (ENRR)[33–43]. However, both the production rate of $NH_3$ and its selectivity is pretty low. Moreover, the underlying mechanism for ENRR is not fully understood. Therefore, the ability to design efficient catalysts with atomic precision offers great opportunities to deepen the mechanistic understanding of ENRR and to further improve their catalytic performance.

To this end, we have devised a facile method for the synthesis of ultrafine bimetallic $Au_4Pt_2$/G SCC containing a partially ligand-protected six-metal-atom (Four Au and two Pt) octahedral cluster anchored on graphene for ENRR. To achieve this, we first developed a synthetic approach for the synthesis of atomically precise $Au_4Pt_2(SR)_8$ clusters using thiol as both the ligand and reducing agent. Interestingly, the subsequent partial ligand removal of $Au_4Pt_2(SR)_8$ via thermal treatment allows each cluster to be anchored at a graphene vacancy site, creating the $Au_4Pt_2$/G SCC with superior catalytic performance for ENRR. The aforementioned synthetic strategy can also be extended to Pd atoms in place of Pt while keeping the cluster framework unchanged. Moreover, it is found that $Au_4Pd_2$/G SCC outperforms $Au_4Pt_2$/G SCC and the majority of ENRR catalysts reported (Supplementary

Fig. 1 and Supplementary Table 1) in terms of maximum $NH_3$ yield and faradic efficiency of ammonia production. This allows us to fine tune the catalytic properties of precisely doped ultrafine bimetallic clusters and understand their structure–property correlations at the atomic level.

## Results

**Synthesis and characterization of clusters.** It is generally recognized that metal ions reduced by a strong reducing agent (e.g., $NaBH_4$) are prone to aggregate and form medium-sized clusters or large-sized nanoparticles. Hence, we expect that a weak reducing agent may be favorable for the synthesis of ultrasmall bimetal clusters. It has also been demonstrated that thiols, the common ligand used in the synthesis of gold clusters, are able to reduce Au(III) to Au(I) due to their low electronegativity[44]. Inspired by this, we developed a new method for the synthesis of ultrafine Au–Pt bimetal clusters by using 2-phenylethanethiol ($HSC_2H_4Ph$) as both the ligand and weak reducing regent (see "Methods" for details). The composition of the as-obtained clusters was determined using high-resolution atmospheric pressure chemical ionization mass spectrometry (APCI-MS) as well as thermogravimetric analysis (TGA). As shown in Fig. 1a, an intense peak at m/z ~2274 is observed, which can be assigned to the molecular ion of the bimetal cluster. TGA shows a total weight loss of 48% at temperature above 600 °C, attributed to the desorption of $-SC_2H_4Ph$ ligands (Supplementary Fig. 2). Based on these results, the composition of the clusters can be readily deduced to be $Au_4Pt_2(SR)_8$ (R represents $C_2H_4Ph$). As-assigned cluster structure is further validated by the excellent agreement between the experimental and calculated isotopic MS patterns of $Au_4Pt_2(SR)_8$ (the inset of Fig. 1a).

We also managed to obtain needle-like yellow single crystals of $Au_4Pt_2(SR)_8$ clusters (Fig. 1b) allowing for accurate structural determination by single-crystal X-ray diffraction (Supplementary Table 2). As shown in Fig. 1c, each cluster consists of a distorted octahedron composed of a plane of four Au with two Pt atoms located at the opposite sides of the Au plane. The octahedron is fully protected by eight thiol ligands wherein eight S−Au and eight S−Pt bonds are formed. Interestingly, each $Au_4Pt_2(SR)_8$ cluster can act as a building block for the crystallization into a 1D polymeric chain-like structure (Fig. 1d). The unit cell contains two interconnected clusters linked by the waist Au atom of each $Au_4Pt_2(SR)_8$ (Supplementary Fig. 3). In addition, the 1D polymeric cluster chain was observed to disassemble into individual clusters (refer to TEM and AFM results below) upon dissolving in organic solvents such as toluene or dichloromethane.

To better understand their electronic properties, we preformed scanning tunneling microscope (STM) imaging and spectroscopy (STS) measurement of individual $Au_4Pt_2(SR)_8$ clusters deposited on a graphite surface. After mild annealing at 70 °C, individual clusters with different orientations can be readily imaged (Fig. 2a). However, upon annealing at 100 °C, isolated clusters were found to aggregate into densely packed monolayer islands (Supplementary Fig. 4), indicating the presence of weak interactions between the cluster and substrate. A representative STS curve acquired over a single cluster shows a wide gap-like feature and several prominent peaks outside the gap attributed to the molecular HOMO and LUMO orbitals as labeled in Fig. 2b. The calculated wave function pattern of these orbitals reveals that the HOMO and LUMO of the cluster mainly consist of contributions from the bimetallic $Au_4Pt_2$ core and S atoms, respectively (Fig. 2c). The HOMO–LUMO gap of each supported single cluster is experimentally determined to be 2.67 eV, in reasonably good agreement with that of the gas-phase cluster (2.82 eV) predicted by density functional theory (DFT) calculations. In addition,

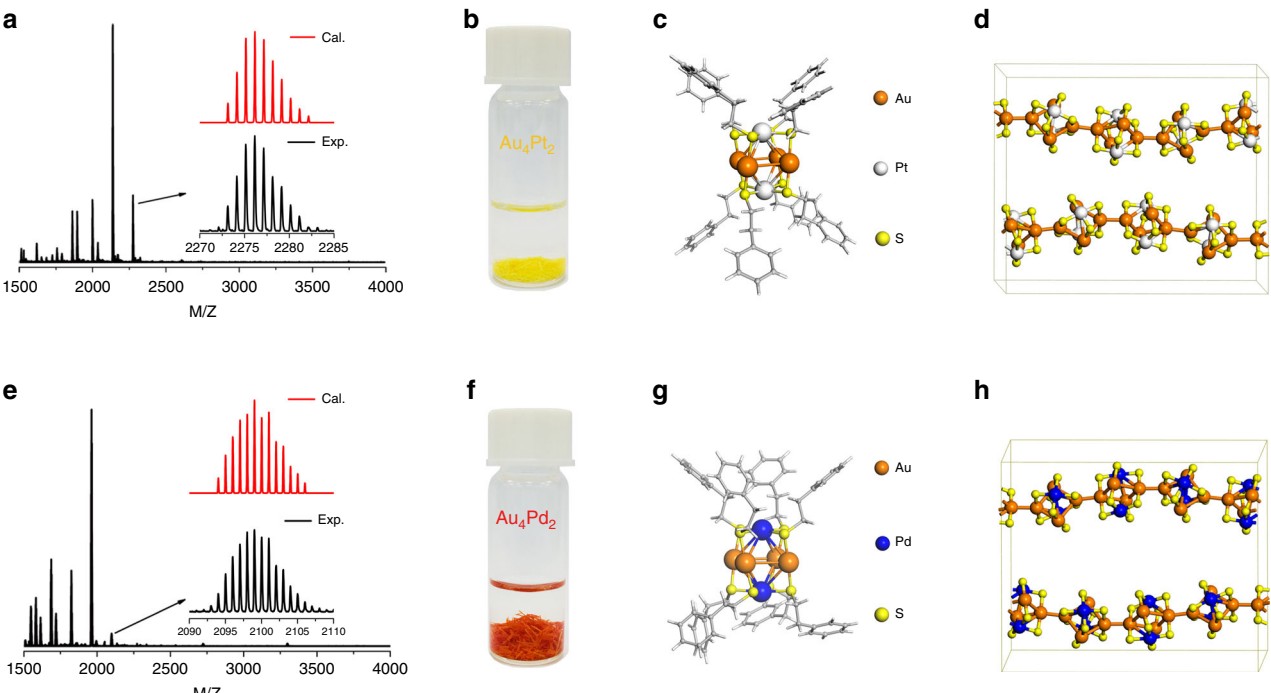

**Fig. 1 Composition and structural characterization of clusters. a**, **e** Mass spectra of $Au_4Pt_2(SR)_8$ and $Au_4Pd_2(SR)_8$. Inset **a** and **e** illustrates the high-resolution experimental and calculated isotopic patterns of $Au_4Pt_2(SR)_8$ and $Au_4Pd_2(SR)_8$, respectively. **b**, **f** Photographic images of the $Au_4Pt_2(SR)_8$ and $Au_4Pd_2(SR)_8$ crystals. **c**, **g** Atomic structures of the clusters and, **d**, **h** their 1D polymeric chains determined by single-crystal X-ray diffraction. Colors: golden, Au; white, Pt; blue, Pd; yellow, S; gray, C. Both C and H atoms are omitted for clarity in (**d** and **h**).

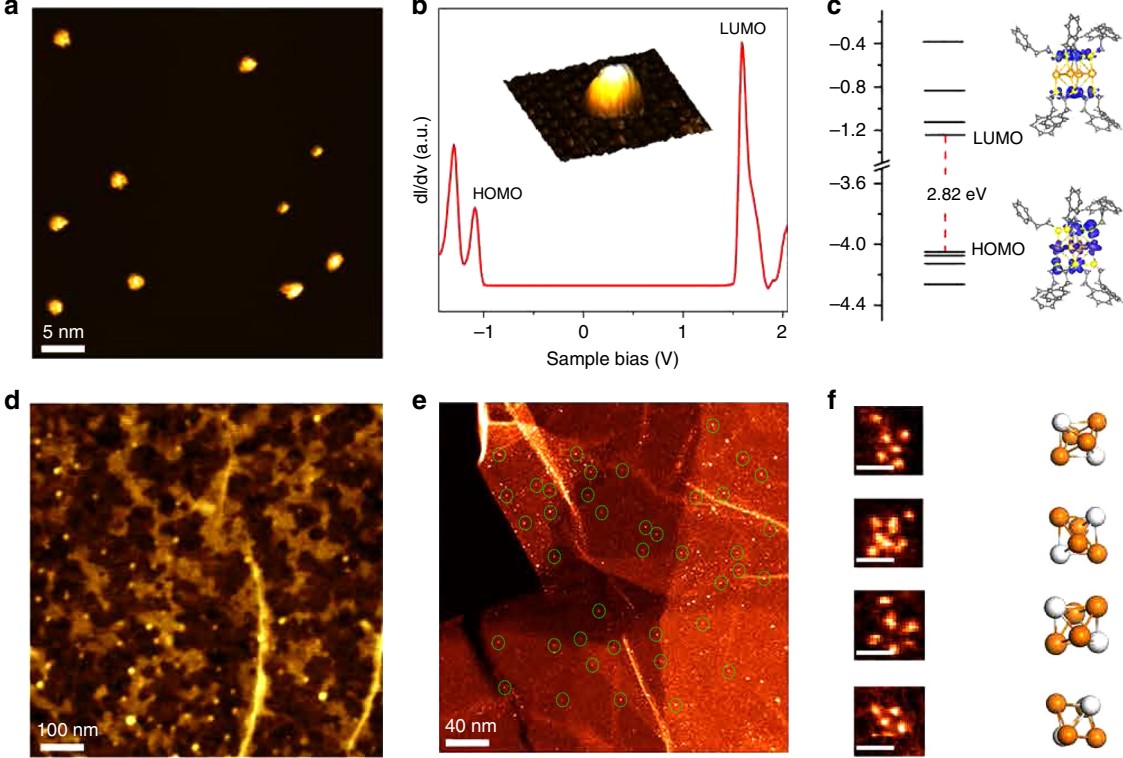

**Fig. 2 Basic characterization of $Au_4Pt_2(SR)_8$. a** Representative STM image of $Au_4Pt_2(SR)_8$ on a HOPG substrate. **b** STS profile of single $Au_4Pt_2(SR)_8$ cluster on graphite surface. **c** Calculated frontier orbitals of single $Au_4Pt_2(SR)_8$ cluster. **d** Representative AFM image of $Au_4Pt_2(SR)_8$/G. **e** STEM-ADF image of $Au_4Pt_2(SR)_8$ deposited on defective graphene. **f** High-resolution STEM-ADF images of single $Au_4Pt_2(SR)_8$ clusters and approximate corresponding orientations(scale bars in Fig. 2f are 5 Å). Colors: golden, Au; white, Pt; yellow, S; gray, C. Sulfur, carbon, and hydrogen atoms are omitted for clarity in (**f**).

atomic force microscope (AFM) imaging shows that a relatively uniform distribution of single $Au_4Pt_2(SR)_8$ clusters can be achieved on high quality monolayer graphene (Fig. 2d). AFM line profile reveals a height of ~2 nm for each bright dot, in line with the expected size of individual clusters (see Supplementary Fig. 5).

All the above-mentioned observations highlight that fully ligand-protected bimetallic clusters retain their structural integrity upon deposition on a weakly interacting substrate, but these clusters lack the desired stability and activity required for catalysis. Hence, we selected defective graphene derived from the reduction of graphene oxide to anchor the $Au_4Pt_2(SR)_8$ clusters for the fabrication of robust and active SCCs for the electrochemical $N_2$ reduction as will be discussed below.

**Synthesis and characterization of $Au_4Pt_2/G$ catalyst for ENRR.** The structural defects in graphene usually act as active sites for reaction with ligand-protected clusters and organic–metal complexes, and eventually bind them via partial removal of organic ligands[45]. Partial ligand removal often reactivates the otherwise inert fully protected metal clusters for catalysis due to alteration of electronic structures[13,46]. For their transformation into stable finely dispersed SCCs, it is most likely that individual $Au_4Pt_2(SR)_8$ clusters were immobilized at the vacancy sites of chemically derived graphene through partial ligand removal. Indeed, we observed monodispersed clusters stabilized on defective graphene (denoted as $Au_4Pt_2/G$) through large-area scanning transmission electron microscope (STEM), as shown in Fig. 2e. Several representative high magnification STEM images (Fig. 2f) also reveal that the majority of bright dots contain a cluster of six atoms, as expected for the bimetallic $Au_4Pt_2$ cluster. We also found that the atomic arrangements of the imaged clusters vary, which can be attributed to the different viewing

direction of the clusters or electron-beam induced cluster dissociation.

We then evaluated the ENRR performance of the as-prepared $Au_4Pt_2/G$ catalyst in comparison with that of $Au_4Pt_2(SR)_8$ without the graphene support using an aqueous-based electrochemical setup as illustrated in Fig. 3a. It was observed that $Au_4Pt_2/G$ SCC demonstrates a significantly higher ENRR activity as compared to $Au_4Pt_2(SR)_8$ at all the applied reduction potentials (Fig. 3b, c and Supplementary Fig. 6). The $Au_4Pt_2(SR)_8$ catalyst demonstrates a maximum $NH_3$ yield of 7.9 µg mg$^{-1}$ h$^{-1}$ (Fig. 3b) with a faradic efficiency (FE) of 9.7% at −0.1 V (Fig. 3c). In contrast, the $Au_4Pt_2/G$ SCC generates a maximum $NH_3$ yield of up to 23.6 µg mg$^{-1}$ h$^{-1}$ at −0.1 V, three times higher than that of the $Au_4Pt_2(SR)_8$ catalyst (Fig. 3b and Supplementary Table 3 and area normalized yield rate see Supplementary Table 4). Hence, this observation suggests that the defective graphene support plays an important role in optimizing the catalytic activity of the bimetal cluster in ENRR. It is worth mentioning that the ENRR performance ($NH_3$ yield and FE) of both the $Au_4Pt_2(SR)_8$ and $Au_4Pt_2/G$ catalysts decline as the reduction potential becomes more negative. This trend can be rationalized by the fact that at more negative potentials, the hydrogen evolution reactions (HER) become dominant, which severely limits ENRR toward $NH_3$ production. We also found that no $NH_3$ can be detected when defective graphene alone was employed as the catalyst for ENRR, or when the same experiment was performed in an argon saturated electrolyte without the $N_2$ source (Supplementary Fig. 6). Nuclear magnetic resonance (NMR) spectroscopy was also employed to determine the generation of ammonia. The 1H resonance coupled to $^{14}N$ in $^{14}NH_4^+$ is split into three symmetric signals with a spacing of 52 Hz (Fig. 3d)[26]. In addition, we conducted an isotopic $^{15}N$ labeling to further confirm the source of nitrogen for $NH_3$ production. A doublet pattern with the coupling constant of $J_{N-H} = 72$ Hz attributed to $^{15}NH_4^+$ was

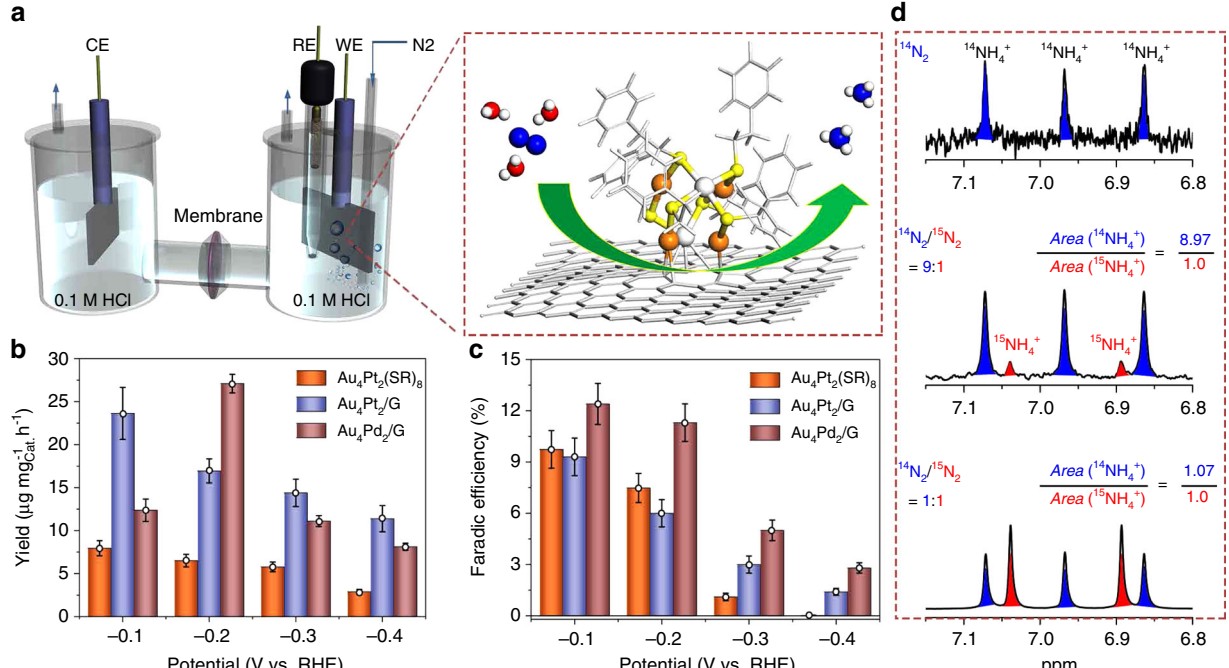

**Fig. 3 Electrochemical $N_2$ reduction reaction using different catalysts. a** Schematic illustration of the experimental setup for ENRR. **b** $NH_3$ production rate. **c** Faradic efficiency of ammonia production at different potentials. **d** $^1$H-NMR spectra of $^{14}NH_4^+$ and $^{15}NH_4^+$ produced from ENRR with different ratios of $^{14}N_2/^{15}N_2$ as initial gas source, including pure $^{14}N_2$, 9:1 and 1:1 as labeled in (**d**). The integrated peak areas associated with $^{14}NH_4^+$ and $^{15}NH_4^+$ are determined to be 9:1 and 1:1, which are proportional to the initial ratio of $^{14}N_2/^{15}N_2$ gas. The error bars in b–c denote standard deviation of three technical replicates.

observed in the $^1$H-NMR spectra (Fig. 3d)[26,47]. In order to validate the yield of NH$_3$, we developed a new approach involving a direct comparison between the ratio of a mixture $^{14}N_2/^{15}N_2$ gas used and the ratio of $^{14}NH_4^+/^{15}NH_4^+$ produced. When a mixture $^{14}N_2$ and $^{15}N_2$ gas with a mole ratio of 9:1 (or 1:1) is used, the ratio of $^{14}NH_4^+$ and $^{15}NH_4^+$ is determined to be 8.97/1 (or 1.07/1) (obtained from NMR signals associated with the $^{14}NH_4^+$ and $^{15}NH_4^+$), proportional to the initial gas ratio of $^{14}N_2/^{15}N_2$ (Fig. 3d). All these results suggest that the NH$_3$ obtained does not originate from the electrolyte and/or materials used in the electrochemical system. The established correlation between the ratio of a mixture $^{14}N_2/^{15}N_2$ gas used and the ratio of $^{14}NH_4^+/$ $^{15}NH_4^+$ validates the NH$_3$ yield determined in our case.

**Probing the origin of catalytic activation of N$_2$.** To gain a deep understanding of the local chemical environment of the active sites, we carried out X-ray absorption fine structure (XAFS) measurements to monitor the change in chemical bonding and oxidation state of the metal species upon anchoring of the bimetallic clusters on graphene[48]. As revealed in Fig. 4a, higher white line intensity was observed in the Pt L$_3$-edge spectrum of Au$_4$Pt$_2$/G as compared to that of Au$_4$Pt$_2$(SR)$_8$, suggesting a higher density of $d$ band holes at the Pt sites of Au$_4$Pt$_2$/G. This can be attributed to charge depletion of the $d$ band due to a strong cluster–substrate interaction[49]. A detailed analysis of Pt L$_3$-edge FT-EXAFS reveals that the Pt–S bond is stretched to 1.78 Å for the anchored Au$_4$Pt$_2$ cluster as compared to that of the unsupported Au$_4$Pt$_2$(SR)$_8$ cluster (1.75 Å) (see Supplementary Fig. 7). In addition, the Au L$_3$-edge XAFS spectrum (Fig. 4b) shows a negligible change of spectroscopic features before and after the anchoring of clusters on graphene. Therefore, it is most likely that the ligand-detached Pt atom is bonded to the carbon atom at the vacancy site via partial ligand removal during the thermal treatment. Such a cluster anchoring process is analogs to the fabrication of surface supported single site molecule catalysts reported previously[45].

In order to determine the atomic structures of the Au$_4$Pt$_2$/G SCCs, we performed DFT calculations with van der Waals corrections (in a D2 format) in combination with a standard simulation of X-ray absorption near edge structure (XANES) (see Fig. 4). Based on our XANES simulation and the plausible surface reaction mechanism, it is highly possible that each cluster undergoes partial ligand removal, leading to a subsequent bonding to carbon atoms at the vacancy of graphene. We hence propose several possible atomic configurations of Au$_4$Pt$_2$/G along this line, which are further optimized via DFT calculations. Our calculations reveal a stable structure consisting of partially ligand-protected Au$_4$Pt$_2$(SR)$_6$ bonded to carbon atoms at graphene vacancy, wherein two ligands at the base of each Au$_4$Pt$_2$(SR)$_8$ cluster are eliminated in order to form Pt–C anchoring bond (as illustrated in Fig. 4c). In addition, we also tested other proposed structures such as graphene-supported Au$_4$Pt$_2$(SR)$_8$ cluster (without any missing ligand, Fig. 4e), Au$_4$Pt$_2$(SR)$_7$, Au$_4$Pt$_2$(SR)$_4$, Au$_4$Pt$_2$(SR)$_2$, and Au$_4$Pt$_2$(SR)$_0$ (missing one, four, six, eight ligands, respectively, refer to Supplementary Figs. 8 and 9). However, all the simulated XANES of these proposed structures do not agree with the experimental XANES data.

To understand how the ligand removal modifies the electronic properties of as-formed cluster in a more intuitive manner, we calculated the detailed energy levels of Kohn–Sham molecular orbitals for both Au$_4$Pt$_2$(SR)$_6$ and Au$_4$Pt$_2$(SR)$_8$ (R=H) using the Amsterdam Density Functional (ADF) program. The removal of two ligands not only reduces the electronic gap of as-formed clusters but also creates two singly occupied electrons derived from 5$d$ and 6$s$ orbitals of Pt and Au, respectively (Fig. 5a). These two energetic electrons on the lower valent metal atoms may facilitate the electron transfer from the cluster to the N$_2$ $\pi^*$ orbitals, resulting in N$_2$ activation. We found that N$_2$ adsorption cannot proceed over any site of Au$_4$Pt$_2$(SR)$_6$ once it is anchored on defect-free graphene (Supplementary Fig. 10). All these results suggest that both Au$_4$Pt$_2$(SR)$_6$ and interfacial defect in graphene play crucial roles not only in the catalyst fabrication but also in the catalytic N$_2$ activation, in agreement with previous studies of graphene-supported metal catalysts[45,50,51].

To probe the origin of catalytic activity of Au$_4$Pt$_2$/G (Note Au$_4$Pt$_2$/G is used to represent the actual structure of Au$_4$Pt$_2$(SR)$_6$/ G for the sake of consistency), we then performed periodic DFT calculations (with D2 correction) to determine the atomic structure of the active site in this system. N$_2$ adsorption is known to be the key step in ENRR. Amongst the various N$_2$ adsorption configurations tested, the most stable one identified is shown in Supplementary Fig. 11. The adsorption energy of N$_2$ for this configuration is estimated to be −0.38 eV, wherein two N atoms are bonded to carbon atoms of graphene and adjacent Pt/Au atoms, respectively. The calculations also reveal that the Fermi energy ($E_F$) of graphene-supported Au$_4$Pt$_2$/G is rebalanced toward the LUMO of N$_2$ (see Fig. 5b), resulting in a small energy separation (~0.6 eV) between $E_F$ and N$_2$ LUMO, consistent with the previous molecular DFT results. This facilitates electron transfer from the active site to the N$_2$ LUMO (Fig. 5b) and the activation of N$_2$. Bader charge analysis (inset of Supplementary Fig. 11 shows the corresponding charge redistribution plot) has shown that the N$_2$ molecule gains a total of 1.44 electrons from the active site of Au$_4$Pt$_2$/G. In addition, the projected density of states (PDOS) shown in Fig. 5c and Supplementary Fig. 12 reveal the detailed electronic interaction between the atoms of the active site and N$_2$. The LUMO of gas-phase N$_2$ consisting of degenerate $p_x$ and $p_z$ orbitals is split into two nondegenerate orbitals upon its adsorption at the active site due to the low local symmetry and different degrees of electronic interaction between the $p_x$ and $p_z$ orbitals of N$_2$ and the Au, Pt, and C orbitals. The HOMO of N$_2$ ($p_y$) is mixed with the $d$ orbitals of Pt/Au, as evidenced from a significantly broadened $p_y$ PDOS upon its adsorption. The strong orbital interaction between both N$_2$ and metal atoms results in a significant electron transfer from the $d$ orbitals of the metal species to the $\pi^*$ antibonding orbitals of N$_2$ in combination with an interesting back-donation mechanism involving a partial electron transfer from the HOMO of N$_2$ ($\sigma$ bonding) back to the metal centers (Fig. 5c and Supplementary Fig. 12). This is analogous to the N$_2$ activation mechanism reported in conventional transition metal catalysts[52].

In addition to N$_2$ activation, we also performed the ground-state calculations with DFT + D2 for possible configurations to estimate the energy profiles of the plausible reaction pathways (Fig. 5d and Supplementary Fig. 13). It is observed that the formation of activated N$_2$* at the catalytic center is energetically favored by 0.38 eV. We then calculated the energy profiles of the two possible reaction pathways for the subsequent protonation of activated N$_2$* species. As shown in pathway I (Fig. 5d), the protonation of N atom bonded graphene occurs first (refer to Supplementary Fig. 13 for details), followed by the protonation of second N atom bonded to the cluster, which leads to the formation of the first and second NH$_3$ molecule respectively. The results also reveal that the rate-limiting step of pathway I is the desorption of the second NH$_3$ in the final reaction step. The reaction barrier of the rate-liming step is estimated to be 0.91 eV, which can be readily surmounted upon the application of an electrochemical potential[38]. The energy profile of the pathway II is shown in Supplementary Fig. 14, in which the protonation of N

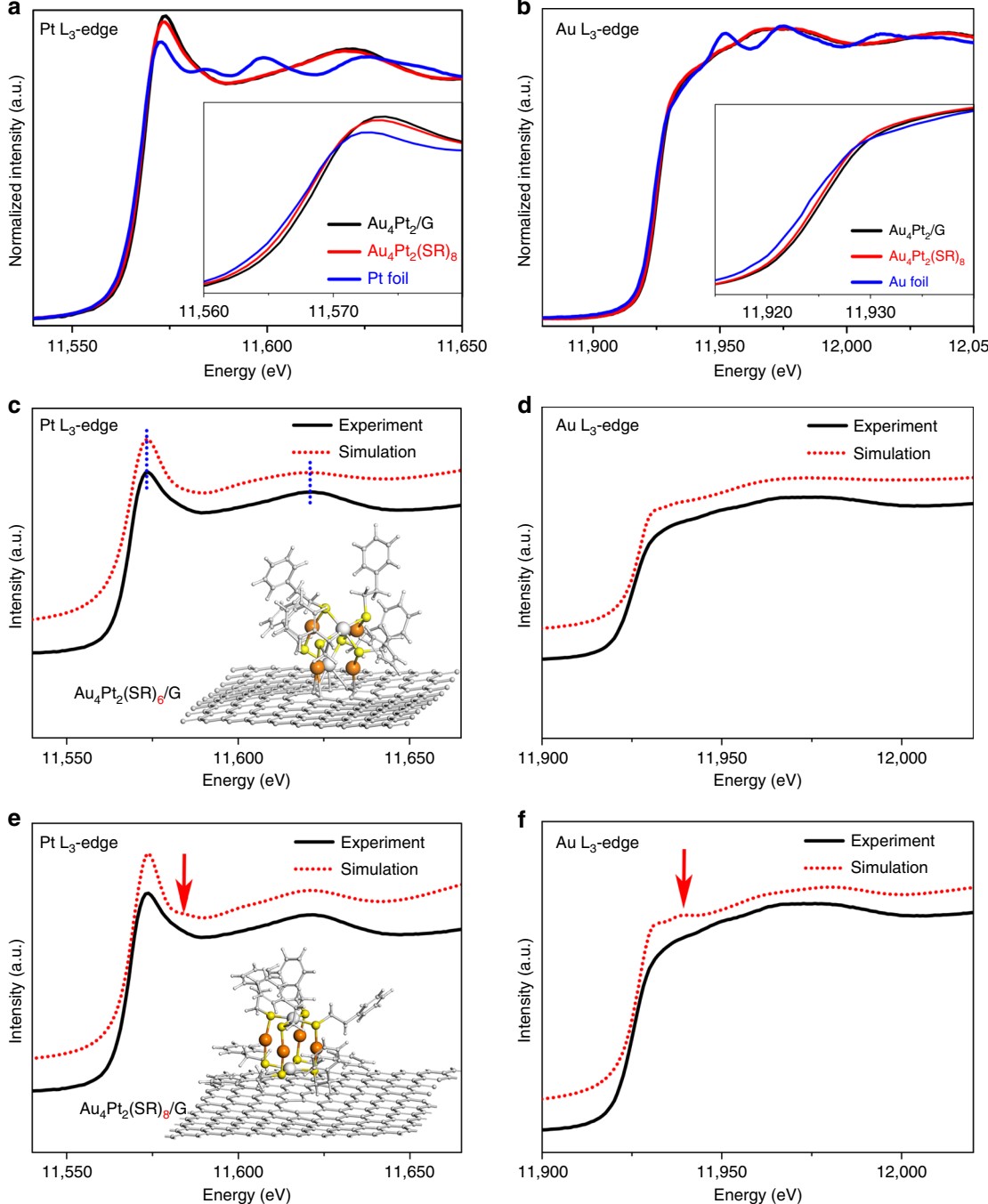

**Fig. 4 Pt and Au L₃-edge XANES spectra. a, b** Normalized XANES spectra of the Pt L₃- and Au L₃-edges for $Au_4Pt_2$/G, $Au_4Pt_2(SR)_8$, and Pt foil (insets show expanded data). **c, d** Comparison of experimental XANES spectra of the Pt L₃- and Au L₃-edges for $Au_4Pt_2$/G (black line) with that of the simulated spectrum (red dot-line) using the DFT-modeled structure, $Au_4Pt_2(SR)_6$/G (inset). **e, f** Comparison of the experimental XANES spectrum of the Pt L₃- and Au L₃-edge of the $Au_4Pt_2$/G (black line) with that of the simulated spectrum (red dot-line) based on the DFT-modeled structure, $Au_4Pt_2(SR)_8$/G (inset).

atom bonded metal cluster occurs first. For this pathway, there are two rate-limiting steps: (1) desorption of the first $NH_3$ with a barrier of 0.74 eV; (2) the other one is the formation of the second $NH_3$ with a high barrier of 2.42 eV. Therefore, our calculation results show that the pathway I is more energetically favorable.

The mechanistic insights into the $N_2$ activation obtained herein motivated us to use Pd in place of Pt as the dopant for the synthesis of a new bimetallic $Au_4Pd_2(SR)_8$ cluster with the same structural framework (Fig. 1e–h and Supplementary Table. 5). This would allow

us to precisely control the doping of SCCs and fine tune its catalytic performance. We were able to successfully synthesize Pd-doped bimetallic clusters $Au_4Pd_2(SR)_8$ on the gram scale using the same synthetic protocol as described earlier (Fig. 1e–h, Supplementary Fig. 15). It was observed that the $Au_4Pd_2$/G catalyst yields a $NH_3$ production rate of 13.1 μg mg⁻¹ h⁻¹ at −0.1 V, lower than that of $Au_4Pt_2$/G catalyst at the same potential. This indicates that $Au_4Pd_2$/G has a lower ENRR activity compared to $Au_4Pt_2$/G. However, we obtained a maximum $NH_3$ yield rate of 27.1 μg mg⁻¹ h⁻¹ with a FE

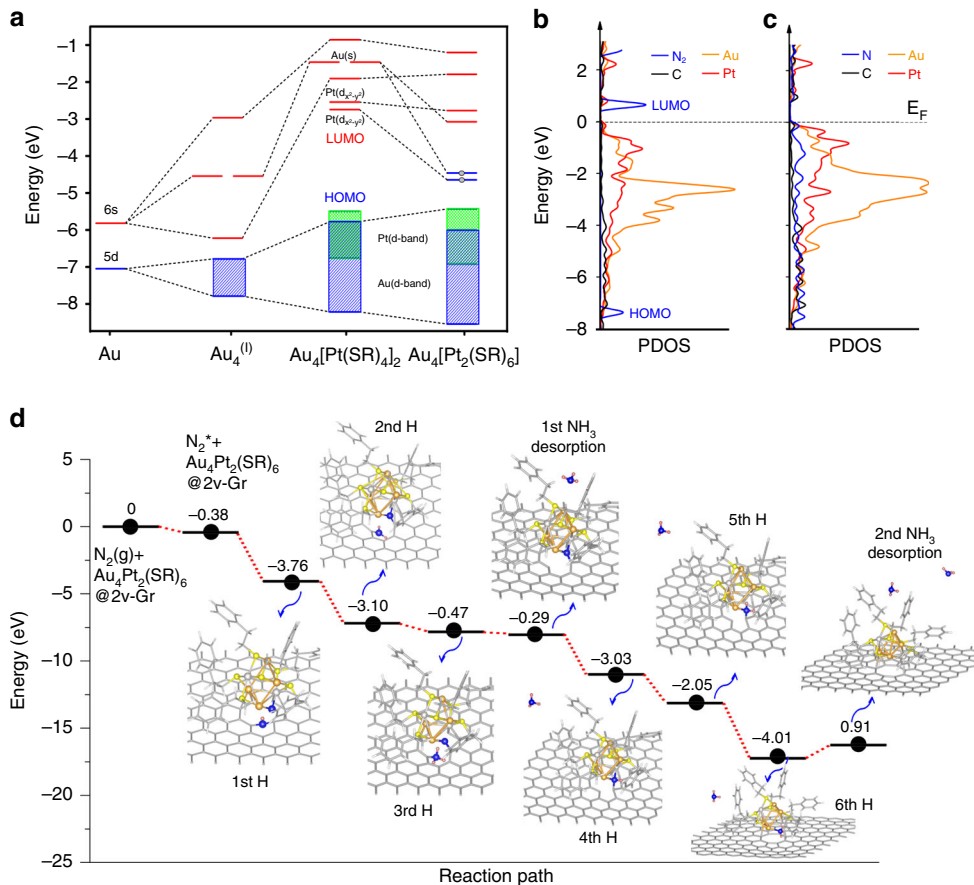

**Fig. 5 Probing the catalytic origin of Au$_4$Pt$_2$/G SCC. a** Schematic energy-level diagram of unsupported Au$_4$Pt$_2$(SR)$_8$ and Au$_4$Pt$_2$(SR)$_6$ clusters. Blue and red lines represent filled and empty orbitals respectively. Blue and green blocks represent the d band of Au and Pt, respectively. The two low-lying electrons are shown as dots. **b, c** Calculated PDOS of Au$_4$Pt$_2$/G catalysts without and with N$_2$ adsorption. **d** Calculated energy profile of the proposed reaction pathway I in which the protonation of N atom bonded with graphene occurs first. Note that NH$_3$ form when the 3rd and 6th H added.

of ~12% at a more negative potential of −0.2 V for Au$_4$Pd$_2$/G, actually outperforming the Au$_4$Pt$_2$/G (Fig. 3b, c). This suggests that HER could be more effectively suppressed in this system as compared to that of Au$_4$Pt$_2$/G, consistent with the HER performance of two bimetallic SCCs tested (Supplementary Fig. 16). We note that pure Au and Pt nanoclusters (e.g. Au$_6$ or Pt$_6$) with the same octahedral framework as that of Au$_4$Pt$_2$ have not been obtained. To further demonstrate the synergistic effect of bimetallic nanoclusters, the ENRR catalytic performance of a pure Au$_{25}$ and Pt nanoclusters (with an average size of 1 nm) was also evaluated. As shown in Supplementary Figs. 17 and 18, the results reveal a poorer ENRR performance of both pure Au and Pt clusters as evidenced by a lower NH$_3$ yield and lower faradic efficiency compared to that of bimetallic Au$_4$Pt$_2$/G and Au$_4$Pd$_2$/G catalysts synthesized. Therefore, these results further confirm that hetero-dopant (Pt or Pd) of bimetallic clusters play important roles in the enhanced catalytic performance of ENRR.

The catalytic cycling stability is another critical parameter of ENRR performance for practical applications. As shown in Supplementary Fig. 19, both ammonia yield and FE remain nearly constant during the multiple cycling tests of both SCCs. Large-area TEM images of both Au$_4$Pt$_2$/G and Au$_4$Pd$_2$/G catalysts show little morphological variation before and after reaction. In addition, STEM images of both catalysts reveal that Au$_4$Pt$_2$ and Au$_4$Pd$_2$ anchored on graphene still contain a cluster of six-atom after ENRR reaction (Supplementary Fig. 20). Moreover, XAFS measurement of the Au, Pt L$_3$-edges and Pd K-edge for both Au$_4$Pt$_2$/G and Au$_4$Pd$_2$/G catalysts shows a negligible spectrum change before and after reactions, which further proves the high cycling stability of both catalysts (Supplementary Fig. 21).

## Discussion

In summary, we have devised a synthetic approach for the synthesis of ultrafine bimetallic Au$_4$Pt$_2$(SR)$_8$ clusters. A sequential anchoring of these bimetallic clusters on defective graphene allows for the synthesis of atomically precise SCC for efficient electrochemical N$_2$ reduction. A nanoscale confined interfacial between the graphene substrate and Au$_4$Pt$_2$(SR)$_6$ cluster acts as the active site for N$_2$ fixation. The heteroatom dopant is found to play an indispensable role in the back donation of electrons from the supported bimetal cluster to the N$_2$ antibonding π*-orbitals, contributing to N$_2$ activation. We also demonstrate that the catalytic properties of the ultrafine bimetallic clusters can be further tuned via precise replacement of the heteroatom dopant. Our findings have opened up a new avenue for the design of atomically precise SCCs with dopant-controlled reactivity for a wide range of industrially important catalysis.

## Methods

**Materials**. All chemicals are commercially available and used as received. In our experiment, we used the ultrapure water (resistivity 18.2 MΩ cm) produced by a Milli-Q NANO water purification system. Tetrachloroauric (III) acid (HAuCl$_4$·3H$_2$O), hydrogen hexachloroplatinate (IV) hexahydrate (Na$_2$PtCl$_6$·6H$_2$O), palladium(II) chloride (PdCl$_2$), tetraoctylammonium bromide (TOABr), 2-phenylethanethiol (PhC$_2$H$_4$SH) were purchased from Sigma-Aldrich. Reduced graphene oxide (G) were purchased from Nanjing XFNANO Materials

Tech Co., Ltd. Tetrahydrofuran (THF), methanol, dichloromethane, petroleum ether, and toluene were purchased from Sinopharm chemical reagent Co., Ltd.

**Synthesis of $Au_4Pt_2(SR)_8$ clusters.** Typically, 305 mg of $HAuCl_4 \cdot 3H_2O$ and 200 of $Na_2PtCl_6 \cdot 6H_2O$ were dissolved in THF (20 mL). Subsequently, tetra-octylammonium bromide (640 mg) was also added into the solution followed by stirring for 5 min After a complete dissolution of all the solid precursors, 830 μL 2-phenylethanethiol was added to the flask followed by an extended stirring for 2 h. The yield of $Au_4Pt_2(SR)_8$ is determined to be ~28 %.

**Synthesis of $Au_4Pd (SR)_8$ clusters.** We adopt the same protocol as described above for the synthesis of $Au_4Pd_2(SR)_8$ clusters. Here we used 68 mg $PdCl_2$ as the precursor for the synthesis of $Au_4Pd_2(SR)_8$ clusters. The yield of $Au_4Pd_2(SR)_8$ is estimated to be ~78 %.

**Single-crystal X-ray diffraction.** The data were collected at 263 K (for $Au_4Pt_2(SR)_8$) and 100 K (for $Au_4Pd_2(SR)_8$) using a four circles goniometer Kappa geometry, Bruker AXS D8 Venture, equipped with a Photon 100 CMOS active pixel sensor detector. A Molybdenum monochromatized ($\lambda = 0.71073$ Å) X-Ray radiation was used for the measurement. Data were corrected for absorption effects using the Multi-Scan method SADABS. The atomic structure of single crystal was solved by direct methods and further refined by full matrix least squares using the SHELXTL 6.1 bundled software package.

**Sample characterizations.** High-resolution APCI-MS was performed on an MicrOTOF-QIImass spectrometer (Bruker) in a positive mode. Compass Iso-topePattern was used to simulate the isotopic pattern. The UV/vis/NIR absorption spectra were measured using a UV-3600 spectrophotometer (Shimadzu) at room temperature. TGA (~3 mg sample used) was conducted in a $N_2$ atmosphere (flow rate ~50 mL/min) at a heating rate of 10 °C/min using a TG/DTA 6300 analyzer. To determinate the loading of clusters on graphene, as-obtained $Au_4Pt_2/G$ or $Au_4Pd_2/G$ (G represents graphene) samples were dissolved in aqua regia and analyzed by inductively coupled plasma mass spectrometry (ICP-MS, Thermo ScientificXseries II). STEM-ADF imaging was carried out in an aberration-corrected JEOL ARM-200F system equipped with a cold field emission gun and an ASCOR probe corrector at 60 kV. The images were collected with a half-angle range from ~85 to 280 mrad, and the convergence semi-angle was set at ~30 mrad.

**XAFS measurements and XANES simulations.** The XANES and the extended X-ray absorption fine structure (EXAFS) measurements of Pt $L_3$ and Au $L_3$ edge were carried out at the XAFCA beamline of the Singapore Synchrotron Light Source (SSLS).The storage ring of SSLS operated at 700 MeV with a beam current of 250 mA. A Si (111) double-crystal monochromator was applied to filter the X-ray beam. Pt and Au foils were used for the energy calibration, and all samples were measured under transmission mode at room temperature. The XAFS data were analyzed using the Demeter software package[53]. The XANES simulated spectra of Pt and Au $L_3$ edges of all the structures predicted by DFT calculation were modeled using a finite difference method implemented by the FDMNES program. The spin–orbit interaction and relativistic effect are included in our calculations. The XAFS measurement of Au $L_3$ and Pd K edges for $Au_4Pd_2/G$ before and after ENRR were measured in transmission mode at beamline 20-BM-B of Advanced Photon Source in Argonne National Laboratory.

**Setup for electrochemical measurements.** The electrochemical reduction of $N_2$ was carried out using a CHI760 electrochemical station with a three-electrode system. A two-compartment glass H-cell was used and connected by a joint separated by a Nafion117 membrane. The saturated calomel electrode (SCE) and Pt foil were used as the reference and counter electrode, respectively.

**Synthesis of $Au_4Pt_2/G$ and $Au_4Pd_2/G$ catalysts.** Twelve milligrams of $Au_4Pt_2(SR)_8$ (or $Au_4Pd_2(SR)_8$ cluster) single crystals was dissolved in 50 mL toluene and stirred for 30 min. Subsquently, a 80 mg of defective graphene was added into the solution rapidly under intense stirring. After 30 min, 500 mL of ethanol was added into the solution rapidly. The black precipitate was collected by filtration and dry at 150 °C in vacuum.

**Preparation of cathode for ENRR.** Typically, 1 mg catalyst (The loading of metal clusters on graphene is 8.5 wt% for $Au_4Pt_2/G$ and 10.5 wt% for $Au_4Pd_2/G$, respectively) and 5 μL Nafion solution (5 wt%) were dispersed in the absolute ethyl alcohol (100 μL) followed by the sonication for 30 min to form a homogeneous ink. Subsequently, the ink was loaded onto a carbon paper with an area of $2 \times 2$ cm$^2$. As-prepared electrode was dried under ambient conditions.

**Calibration of the reference electrode.** We used a SCE as the reference electrode in all measurements. The reference electrode was calibrated with respect to a reversible hydrogen electrode (RHE). The calibration was performed in the high purity hydrogen saturated electrolyte using Pt foils as both working and

counter electrodes (0.1 M HCl electrolyte). Cyclicvoltammetry measurements were performed at a scan rate of 1 mV s$^{-1}$. The average value of the two potentials at which the $H_2$ oxidation/evolution curves cross at $I = 0$ was treated as the thermodynamic potential for the hydrogen electrode reactions. Therefore, the calibration of the reference electrode in 0.1 M HCl can be obtained using this equation: E (RHE) = E (SCE) + 0.32 V (Supplementary Fig. 22)

**ENRR measurements.** Prior to the test of ENRR, Nafion117 membrane was immersed in 5% $H_2O_2$ aqueous solution at 80 °C for 1 h. Subsequently, the membrane was soaked in ultrapure water at 80 °C for another 1 h. ENRR was performed in a three-electrode configuration consisting of the working electrode (either $Au_4M_2(SR)_8$ or $Au_4M_2/G$ (M = Pt, Pd)), Pt foil counter electrode and SCE reference electrode, respectively. A two-compartment H-shape cell separated by Nafion117 membrane was used for ENRR (Fig. 3a in main text). All the glasswares were first boiled in 0.1 M NaOH for 2 h and washed with ultrapure water. After that, they were boiled in 0.1 M HCl for another 2 h and rinsed at least three times in ultrapure water followed by the vacuum drying for 6 h at 110 °C. In this work, all potentials were converted to the RHE scale. The potentiostatic test for ENRR was conducted in the 0.1 M HCl solution (30 ml) saturated by $N_2$. $N_2$ gas (99.999% purity) was continuously fed to the cathodic compartment during the whole ENRR. The performance of catalysts was evaluated under a controlled potential electrolysis in an electrolyte for 1 h at room temperature. Prior to each electrolysis, the electrolyte was presaturated with $N_2$ via gas bubbling for 30 min. During each electrolysis, the electrolyte was continuously bubbled with $N_2$ at a flow rate of 10 sccm. In addition, the control experiments including the potentiostatic test using (1) 0.1 M HCl solution saturated by argon gas and (2) bare graphene without cluster as catalyst were performed in the same setting. Preliminary purification of gases utilized in the experiments, including pure $^{14}N_2$, a mixture of $^{14}N_2$ and $^{15}N_2$, and Ar have been done before the introduction of them into electrochemical cell. The gases were further purified via the flow through a series of solutions including 1 M NaOH solution, ultrapure water, a concentrated $H_2SO_4$, and ultrapure water, to mitigate the contribution of extrinsic contaminants.

**Determination of ammonia.** The concentration of as-produced ammonia was determined using a modified indophenol blue method[54]. First, 2 mL electrolyte obtained from the electrochemical reaction vessel was added into the 1 M NaOH solution (2 mL) containing salicylic acid and sodium citrate. Second, 1 mL of 0.05 M NaClO and 0.2 mL of 1 wt% $C_5FeN_6Na_2O$ (sodium nitroferricyanide) were added into the above-mentioned solution, which was kept at room temperature for 2 h before the subsequent UV–Vis spectroscopy measurements. We measured the UV–Vis absorbance (at the maximum wavelength of 656 nm) of a series of standard ammonia chloride solutions to prepare the calibration curves for the determination of the ammonia concentration of unknow solutions. The fitting curve reveals a linear relationship between the absorbance and $NH_3$ concentration ($y = 0.429x - 0.015$, $R^2 = 0.999$, see Supplementary Fig. 23).

**The calculation of FE.** The FE is calculated as follows.

$$FE = 3F \times n_{NH_3}/Q, \qquad (1)$$

where $F$ is the Faraday constant (96485 C mol$^{-1}$). $Q$ is the total charge passed through the electrode.

The mole of ammonia ($nNH_3$) was calculated using the following equation:

$$n_{NH_3} = n_{NH_4Cl} = (C_{NH_4Cl} \times V) \times 10^{-6}/M_{NH_4Cl} \qquad (2)$$

Note: $C_{NH_4Cl}$ (μg mL$^{-1}$) refers to the measured $NH_4Cl$ concentration, V (mL) is the volume of the electrolyte (30 mL), $M_{NH_4Cl}$ is the molecular weight of $NH_4Cl$.

**$^{15}N_2$ isotope labeling experiment.** A mixture of $^{14}N_2$ and $^{15}N_2$ (with mole ratios of 9:1 and 1:1, respectively) was used as the feeding gas for the isotopic labeling experiment. The detailed procedure is largely similar to that of $^{14}N_2$ electrochemical experiment despite of minor differences. Before introducing $^{15}N_2$ labeling gas, Ar gas flows through the whole setup for 30 min to remove $^{14}N_2$ and other possible gas inpurities. After purging with sufficient Ar, a mixed gas ($^{14}N_2$ and $^{15}N_2$) with well-defined ratios ($^{14}N_2/^{15}N_2$) is introduced into ENRR system for 20 min with a flow rate of 10 sccm. To generate adequate amount of products for the subsequent NMR analysis, we run the reaction for 10 h. The electrolyte after ENNR was further condensed prior to the 1H-NMR spectroscopy measurement (500 MHz, DMSO-d6).

**Determination of hydrazine.** The concentration of the hydrazine presented in the electrolyte was estimated using a modified method developed by Watt and Chrisp[55]. Supplementary Fig. 24.

**Molecular and periodic DFT calculations.** In the periodic DFT calculations, the geometries of $Au_4Pt_2(SR)_n$ and $Au_4Pd_2(SR)_n$ ($n = 8, 6$) metal clusters are adopted from experimental results and then fully optimized using DFT calculations with $20 \times 20 \times 20$ Å$^3$ supercell. The graphene vacancy structure is modeled by removing two carbon atoms in a supercell with $10 \times 10$ graphene pristine cells including a

25 Å vacuum layer so that the supercell is large enough to contain metal clusters. In all calculations except for energy levels presented in Fig. 5a, Vienna ab-initio Simulation Package is utilized with spin polarization Kohn–Sham formalism[56,57]. The generalized gradient approximation (GGA) in the Perdew–Burke–Ernzerh (PBE) format, with scalar relativistic (SR) effects of Au considered[58], the projector-augmented wave method[59] and a plane wave basis with the cut-off energy of 400 eV are employed in all the calculations. Van der Waals force (through DFT + D2) is also considered. The convergence criteria for electronic steps and structural relaxations were set to $10^{-5}$ eV and 0.01 eV/Å, respectively.

In the molecular DFT calculations of the energy levels of the $Au_4Pt_2(SR)_n$ ($n =$ 8, 6) metal clusters, as presented in Fig. 5a, relativistic DFT quantum chemical methods are adopted as implemented in the ADF (2016.101) program[60–62]. The GGA with the PBE exchange-correlation functional[63] was used, together with the uncontracted TZ2P Slater basis sets for all atoms[64]. Frozen core approximations were applied to the inner shells $[1s^2-2p^6]$ for S and $[1s^2-5d^{10}]$ for Au and Pt atoms. The SR effects were considered by the zero-order-regular approximation to account for the mass–velocity and Darwin effects[65]. In calculations, simplified SR (R=H, $CH_3$) group was used as a substitute of $SCH_2CH_2C_6H_5$ ligand to form the model clusters and to save time for the calculations. As the results are qualitatively similar, we only present the results with R=H here. As the experimental structure of $Au_4Pt_2(SCH_2CH_2C_6H_5)_8$ cluster shows a skeleton with point-group symmetry close to $D_4$, we used $D_4$ symmetry to optimize the simplified model to better understand the electronic structure of the cluster. The stability and reactivity of $Au_4Pt_2(SR)_6$ are simply evaluated by removing two adjacent SR ligands coordinated with the same Pt atom in the unrelaxed cluster.

## Data availability

The X-ray crystallographic coordinates for structures reported in this work have been deposited at the Cambridge Crystallographic Data Center (CCDC) under deposition numbers 2012798 and 2012572 for the polymeric $Au_4Pt_2(SR)_8$ and $Au_4Pd_2(SR)_8$ clusters, respectively. These data can be obtained free of charge from the CCDC via https://www.ccdc.cam.ac.uk/structures/. Check cif files for $Au_4Pt_2(SR)_8$ and $Au_4Pd_2(SR)_8$ polymers are given as Supplementary Dataset 1 and 2, respectively. All other data that support the findings of this study are available from the corresponding author upon reasonable request.

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

## Acknowledgements

Jiong Lu acknowledges the support from MOE grants (MOE2017-T2-1-056 and R-143-000-B47-114) and NUS Flagship Green Energy Program (R-143-000-A55-646). Chuanhao Yao acknowledges financial support from Natural Science Foundation of China (21601193, 21703143), the Fundamental Research Funds for the Central Universities (31020190QD013). Chun Zhang thanks the support of NUS green energy program (R-143-000-A63-114) and Tier 1 (R-144-000-410-114) and also NUS graphene center computer clusters. Jun Li thanks the support from the National Natural Science Foundation of China (21590792, 91645203, and 21521091). The calculations were partially performed by using the supercomputers at Tsinghua National Laboratory for Information Science and Technology and the Center for Computational Science and Engineering (SUSTech). Shibo Xi and Yonghua Du thank the support from the XAFCA beamline of Singapore Synchrotron Light Source and the National Supercomputing Centre, Singapore (https://www.nscc.sg). This research also partially used resources of the Advanced Photon Source, an Office of Science User Facility operated for the U.S. Department of Energy (DOE) Office of Science by Argonne National Laboratory and was supported by the U.S. DOE under Contract no. DE-AC02-06CH11357, and the Canadian Light Source and its funding partners.

## Author contributions

J.L. and C.Y. conceived and designed the project. C.Y. synthesized the bimetallic clusters and prepared the catalysts. C.Y. and W.L. performed the activity test with the assistance of Z.C., H.Y., P.L., C.C., H.X., X.P., X.L., B.L., and C.S. Theoretical calculations were performed by N.G., C.Z., C.X. and Ju. L., S.X., Y.D., and C.J.S. performed the XAFS measurement and XANES simulation. X.Z. and S.J.P. performed the STEM-ADF characterization. Jin. L., H.F., J.S., Z.Q. performed AFM and STM characterization. C.Y., C.Z., and J.L. wrote the manuscript. All authors participated in the discussion of the data and editing the manuscript.

## Competing interests

The authors declare no competing interests.
