## [Peer Review File · Nature Communications]

Reviewers' comments:

Reviewer #2 (Remarks to the Author):

This manuscript reported the synthesis of Au₄Pt₂ clusters on graphene and their applications in NRR. A high faradic efficiency of ~10% was observed. DFT calculations showed that the strong electronic interaction between the clusters and N₂ molecules. This manuscript is not recommended for publication in the present form. It can be reconsidered if the following can be addressed:

- 1) The faradic efficiency is low compared with previous reports on different catalysts using the similar testing methods, i.e., liquid cell.
- 2) The long-term stability of the catalysts needs to be tested.
- 3) The utilization of HCl as the electrolyte should be justified.
- 4) Can the authors test the performance of pure Au or Pt clusters?

Reviewer #3 (Remarks to the Author):

In this study, the authors presented a strategy for the synthesis of cluster catalysts consisting of partially ligand-enveloped Au₄Pt₂ and Au₄Pd₂ clusters on graphene. And the suggested nanoscale confined interfacial space between the graphene substrate and bimetallic cluster is regarded as the active site for N₂ fixation. Although authors design atomically precise bimetallic cluster through simple experimental operations, some indispensable experiments and reasonable explanations are missing at this stage. To be fair, the electrochemical synthesis of NH₃ is a burgeoning research field, thus a rigorous refereeing is needed. Based on the overall evaluation of the article, the manuscript did not meet the standard of Nat. Comm., thus the manuscript should be rejected at this stage.

1. The authors should examine the reducing reagent for the synthesis of ultrafine Au-Pt bimetal clusters by using 2-phenylethanethiol (Page 5) or thiol (Page 4).
2. The authors described "...an intense peak at m/z ~2274 is observed, which can be assigned to the molecular ion of the bimetal cluster..." (Page 5). A detailed analysis should be performed.
3. The authors addressed "...Au₄Pd₂/G SCC outperforms Au₄Pt₂/G SCC...in terms of maximum NH₃ yield and faradic efficiency of ammonia production..." (Page 5). However, the structure characterization is mainly on Au₄Pt₂/G, and the author should do systematic analysis of Au₄Pd₂/G and summarize the reasons for excellent performance.
4. The authors addressed "the 1D polymeric cluster chain was observed to disassemble into individual clusters upon dissolving in organic solvents" (Page 6). However, the experimental evidence is missing.
5. On Page 3 (Supplementary Information), "The yield of Au₄Pt₂(SR)₈ is determined to be ~28%", please describe the yield calculation method in detail.
6. On Page 3 (Supplementary Information), the data of single-crystal X-ray diffraction was collected at 263 K (for Au₄Pt₂(SR)₈) and 100 K (for Au₄Pd₂(SR)₈), respectively. Why collect single-crystal X-ray diffraction data at different temperatures?
7. On Page 9 (Supplementary Information), the thermogravimetric analysis result of Au₄Pt₂(SR)₈

crystals. The curve showed two thermal decomposition processes, and the related reaction process should be answered.

8. In figure 2(f), it is difficult to distinguish the arrangements of Au or Pt atoms (the plotting scale notes are missing), some high-resolution pictures should be provided.

9. In Figure 6 (Supplementary Information), what is the concentration ratio of $^{14}\text{NH}_4^+$ and $^{15}\text{NH}_4^+$ in NMR, whether the value is proportional to the input amount of $^{14}\text{N}_2$ and $^{15}\text{N}_2$ molecules.

10. The ammonia yields and Faradic efficiencies of multiple cycling tests should be tested.

11. The structural characterization of bimetallic cluster after electrochemical N_2 reduction reaction should be provided.

12. In figure 4(c)~(f), the derivative structure of partial ligand removed $\text{Au}_4\text{Pt}_2(\text{SR})_8/\text{G}$ is incomplete at this stage, how about the comparison of experimental XANES spectra with the simulated spectra of $\text{Au}_4\text{Pt}_2(\text{SR})_4/\text{G}$, $\text{Au}_4\text{Pt}_2(\text{SR})_2/\text{G}$ and $\text{Au}_4\text{Pt}_2/\text{G}$.

13. The solvation effect could affect the barrier or free energy due to the H-bond between O atom of H_2O solvation and H atom of NRR intermediates. Thus, the authors should compare this effect on the calculation results.

14. In this study, the N_2 adsorption step is favored over the catalysts, while the desorption process of $^*\text{N}$ is not provided, and subsequent steps should be added in Figure 5d.

15. The PDOS of different orbitals for N_2 adsorbed over the catalyst has been discussed in the manuscript. As an important step, the PDOS results of catalysts with $^*\text{NNH}$ should also be added in the revised manuscript.

Point-by-point response letter

Reviewer #2 (Remarks to the Author):

This manuscript reported the synthesis of Au₄Pt₂ clusters on graphene and their applications in NRR. A high faradic efficiency of ~10% was observed. DFT calculations showed that the strong electronic interaction between the clusters and N₂ molecules. This manuscript is not recommended for publication in the present form. It can be reconsidered if the following can be addressed:

Comment 1. *The faradic efficiency is low compared with previous reports on different catalysts using the similar testing methods, ie., liquid cell.*

Response: We thank the referee for your valuable comments.

Typically, several parameters need to be taken into account for the evaluation of the overall catalytic performance for NRR. These include the NH₃ yield, faradic efficiency (FE) and the potential applied in electrochemical reaction. The optimized catalyst shall offer a high NH₃ yield with a high FE acquired at a low overpotential. In order to have a fair comparison, we need to consider all these factors to evaluate the overall catalytic performance of different catalysts.

We agree with referee that the FE value (~10% for Au₄Pt₂/G and ~12% for Au₄Pd₂/G) achieved in our experiment is not the highest one amongst all the reported values. However, these FE numbers are still considerably good compared to a majority of catalysts reported (refer to **Figure R1 and Table R1**). Moreover, we also need to consider other two factors including maximal NH₃ production rate and the corresponding electrochemical potentials applied.

The ammonia production rate of Au₄Pt₂/G and Au₄Pd₂/G are 23.6 $\mu\text{g} \cdot \text{h}^{-1} \cdot \text{mg}_{\text{cat}}^{-1}$ and 27.1 $\mu\text{g} \cdot \text{h}^{-1} \cdot \text{mg}_{\text{cat}}^{-1}$, respectively, higher than a vast majority of existing electrocatalysts (refer to **Figure R1 and Table R1**).

Additionally, the best catalytic performance of these single cluster catalysts in terms of NH₃ yield and FE can be obtained at very low overpotential (-0.1 and -0.2 V versus RHE for Au₄Pt₂/G and Au₄Pd₂/G respectively).

Therefore, the overall catalytic performance of both Au₄Pt₂/G and Au₄Pd₂/G catalysts reported in this work is superior. Moreover, this work demonstrates that the heteroatom dopant plays an indispensable role in the activation of N₂. Besides the heteroatom Pt, the catalytic performance of bimetallic SCCs can be further tuned by using Pd in place of Pt as the dopant. Our findings

offer a new route for the design of novel SCCs with atomic precision for electrochemical dinitrogen reduction.

Figure R1 (Supplementary Fig. 1). NRR catalytic performance of the Au₄Pd₂/G and Au₄Pt₂/G catalysts in comparison with the catalysts reported.

Table R1 (Supplementary Table 1). NRR catalytic performance of the Au₄Pd₂/G and Au₄Pt₂/G catalysts in comparison with the catalysts reported.

Entry	Potential (V)	NH ₃ yield rate ($\mu\text{g} \cdot \text{h}^{-1} \cdot \text{mg}_{\text{cat.}}^{-1}$)	Faradic efficiency (FE, %)	References
1	0.0	0.213	1.96	Acs. Sus. Chem. Eng., 2017, 5, 10986
2	0.0	7.6	56.5	Nat. Commun., 2019, 10, 341
3	-0.1	4.5	8.2	Nat. Commun., 2018, 9, 1795
4	-0.2	8.3	10.10	Adv. Mater. 2017, 29, 1700001
5	-0.2	23.88	0.217	J. Mater. Chem. A, 2018, 6, 3211
6	-0.2	23.21	10.16	Angew. Chem. Int. Ed. 2018, 57, 6073
7	-0.2	34.83	9.28	ACS Catal., 2019, 9, 336

8	-0.2	21.4	8.11	Adv. Mater., 2017, 29, 1606550
9	-0.2	8.09	11.59	Angew. Chem. Int. Ed. 2018, 57, 10246
10	-0.3	11.3	7.8	Adv. Mater., 2018, 30, 1803694
11	-0.3	34.0	14.6	Angew. Chem. Int. Ed. 2019, 58, 2321
12	-0.3	7.75	13.79	Small 2019, 15, 1805029
13	-0.4	20.4	9.3	J. Mater. Chem. A, 2018, 6, 24031
14	-0.4	29.28	8.34	Adv. Energy Mater. 2018, 8, 1801357
15	-0.5	29.43	0.75	J. Mater. Chem. A, 2018, 6, 12974
16	-0.5	8.6	10.04	Small Methods 2018, 1800333
17	-0.55	43.6	9.26	Nano Energy 2018, 52, 264
18	-0.7	23.32	6.7	Chem. Commun., 2018, 54, 11332
19	-0.7	16.22	1.84	Nanoscale Adv., 2019, 1, 961
20	-0.7	31.37	3.09	Angew. Chem. Int. Ed. 2019, 58, 261
21	-0.75	26.57	15.95	Nat. Commun., 2018, 9, 3485
22	-0.75	28.13	8.56	Chem. Commun., 2018, 54, 12848
23	-0.8	15.9	0.94	ChemCatChem 2018, 10, 4530
24	-0.8	17.04	4.76	Electrochimica Acta 2019, 298, 106
25	-0.9	23.8	< 1.5	ACS Catal. 2018, 8, 1186
26	-0.9	15.13	3.3	J. Mater. Chem. A, 2018, 6, 17303
Au₄Pd₂/G	-0.2	27.1	12.3	This work
Au₄Pt₂/G	-0.1	23.6	9.7	This work

Action: We have placed Figure R1 and Table R1 with the corresponding description in the revised Supplementary information.

Comment 2. The long-term stability of the catalysts needs to be tested.

Response: This is a good point. In view of referee's suggestion, we have performed the long-term stability test of both Au₄Pt₂/G and Au₄Pd₂/G catalysts for NRR.

As shown in **Figure. R2**, the results reveal that both ammonia yield and FE remain nearly constant during the multiple cycling stability tests.

Figure R2 (Supplementary Fig. 20). Long-term stability of both Au₄Pd₂/G and Au₄Pt₂/G catalysts for NRR.

Action: we have included the stability test results in the revised supplementary information.

Comment 3. *The utilization of HCl as the electrolyte should be justified.*

Response: Different types of electrolytes including acidic, basic or buffer solutions have been used for the electrochemical N₂ reduction. HCl has been widely used as a common electrolyte for NRR in the literature (see **Table R2** below).

When HCl is used as the electrolyte, Cl⁻ anion will be oxidized into Cl₂ at the counter electrode. In contrast, O₂ will be generated for other types of electrolytes such as H₂SO₄, KOH, Na₂SO₄, LiClO₄. The mass diffusion and electron transfer for the Cl₂ evolution is faster compared to the OER with the sluggish kinetics. Therefore, HCl electrolyte is preferred because it helps to eliminate the kinetics limitation from the counter electrode.

In addition, we also conducted the control experiment using different electrolytes such as H₂SO₄ or KOH solution. Our results reveal that both the NH₃ yield and FE value show negligible variation when these electrolytes are used in electrochemical N₂ reaction, see **Figure R3**.

Table R2. A list of electrolytes reported for the electrochemical NRR

Category	Electrolytes	References
Acid	0.1 M HCl	Adv. Mater. 2017, 1700001
		Adv. Mater. 2017, 1606550
		Angew. Chem. Int. Ed. 2018, 57, 10246
		Nat. Commun., 2018, 9, 3485
		Angew. Chem. Int. Ed. 2019, 58, 2612
	0.05 M H ₂ SO ₄	Adv. Mater. 2018, 1803498
ACS Catal. 2018, 8, 1186		

Base	0.1 M KOH	Nat. Commun., 2019, 10, 341
Salt	0.1 M Na ₂ SO ₄	Adv. Energy Mater. 2018, 8, 1801357
	0.5 m Li ₂ SO ₄	Adv. Mater. 2018, 30, 1803694
	0.5 M LiClO ₄	Chem. Commun., 2018, 54, 11332
Buffer	Phosphate buffer solution	Nat. Commun., 2018, 9, 1795

Figure R3. The yield and faradic efficiency (FE) of Au₄Pd₂/G catalyst obtained in different electrolytes. Note that all the data points were collected at -0.2 V vs RHE.

Comment 4. Can the authors test the performance of pure Au or Pt clusters?

Response: We thank the reviewer for your suggestion. Two bimetallic clusters synthesized in this work (Au₄Pt₂ and Au₄Pd₂) share the same octahedral framework. This is crucial to establish the composition-property correlation by a direct comparison of their electrochemical NRR performance. Unfortunately, pure Au or Pt clusters (e.g. Au₆ or Pt₆) with the same octahedral framework as that of Au₄Pt₂ have not been obtained up to date. In light of referee's comments,

we have tested the NRR performance of Au₂₅ and a newly prepared Pt cluster (with an average size of 1 nm). The results (**Figure R4**) reveal a poorer NRR performance of both pure Au and Pt clusters as evidenced by a lower yield and lower faradic efficiency compared to that of Au₄Pt₂ and Au₄Pd₂. Therefore, these results further confirm that hetero-dopant (Pt or Pd) of bimetallic clusters play importance roles in the enhanced catalytic performance of NRR.

Figure R4 (Supplementary Fig. 18) TEM images of pure Au and Pt cluster catalysts (note that both Au and Pt clusters are also anchored on the defective graphene)

Figure R5 (Supplementary Fig. 19). Catalytic performance of pure Au and Pt clusters catalysts in ENRR.

Action: We have placed Figure R4 and Figure R5 with the corresponding description in the revised Supplementary information.

Reviewer #3 (Remarks to the Author):

In this study, the authors presented a strategy for the synthesis of cluster catalysts consisting of partially ligand-enveloped Au₄Pt₂ and Au₄Pd₂ clusters on graphene. And the suggested nanoscale confined interfacial space between the graphene substrate and bimetallic cluster is regarded as the active site for N₂ fixation. Although authors design atomically precise bimetallic cluster through simple experimental operations, some indispensable experiments and reasonable explanations are missing at this stage. To be fair, the electrochemical synthesis of NH₃ is a burgeoning research field, thus a rigorous refereeing is needed. Based on the overall evaluation of the article, the manuscript did not meet the standard of Nat. Comm., thus the manuscript should be rejected at this stage.

Comment 1. The authors should examine the reducing reagent for the synthesis of ultrafine Au-Pt bimetal clusters by using 2-phenylethanethiol (Page 5) or thiol (Page 4).

Response: We apologize for the confusion and would like to clarify this point as follows.

2-Phenylethanethiol is considered as one specific type of thiol. In page 4, we use “thiol” because it refers to a general concept. In page 5, we specify the thiol used in our work is *2-phenylethanethiol*. We would like to point out that only *2-phenylethanethiol* is used as the reducing reagent in our work.

Comment 2. The authors described “...an intense peak at $m/z \sim 2274$ is observed, which can be assigned to the molecular ion of the bimetal cluster...” (Page 5). A detailed analysis should be performed.

Response: We appreciate the reviewer’s suggestion. In view of referee’s comments, we have provided a detailed analysis of MS spectrum in our revised supporting information.

Mass spectroscopy (MS) and TGA are two commonly-used characterization techniques for metal clusters. High resolution MS provides the accurate analysis of molecular mass of molecular-like clusters. TGA will offer additional information related to the mass contribution of ligand involved in the cluster.

The molecular weight of as-prepared Au₄Pt₂(SC₂H₄Ph)₈ cluster is determined to be 2274 Da (the peak with the largest M/Z value as shown in Figure 1a).

The mass contribution of ligand is determined to be 48% based on the TGA result. Therefore, the mass of ligand can be calculated as $2274 \times 48\% = 1091.5$ Da. The total number of ligand can be

calculated as $1091.5/137(\text{the molecular weight of } \text{SC}_2\text{H}_4\text{Ph}) = 7.97$, in consistent with the total eight $\text{SC}_2\text{H}_4\text{Ph}$ ligands for each cluster (Supplementary Fig. 2).

Based on the aforementioned analysis, the molecular composition of the cluster can be deduced as $\text{Au}_4\text{Pt}_2(\text{SC}_2\text{H}_4\text{Ph})_8$, which is further corroborated by the theoretical isotopic MS patterns of $\text{Au}_4\text{Pt}_2(\text{SC}_2\text{H}_4\text{Ph})_8$ (the inset of Fig. 1a).

Actions: we have included a detailed analysis in the revised Supplementary information.

Comment 3. *The authors addressed “...Au4Pd2/G SCC outperforms Au4Pt2/G SCC...in terms of maximum NH3 yield and faradic efficiency of ammonia production...” (Page 5). However, the structure characterization is mainly on Au4Pt2/G, and the author should systematic analysis of Au4Pd2/G and summarize the reasons for excellent performance.*

Response: This is a good point. In view of referee’s comment, we carried out additional experiment to characterize the properties of $\text{Au}_4\text{Pd}_2/\text{G}$ catalyst before and after reaction using a set of characterization techniques such as TEM, STEM and XAFS. The results with the corresponding description are included in the revised Supporting information.

Figure R6 (Supplementary Fig. 21). Structural characterization of bimetallic clusters before and after electrochemical N₂ reduction. a, b TEM images of Au₄Pt₂/G before and after ENRR. c, d TEM images of Au₄Pd₂/G before and after ENRR. (Inset b and d are spherical aberration corrected STEM images of Au₄Pt₂/G and Au₄Pd₂/G catalysts after ENRR. Scale bars for inset are 5Å)

Action: We have placed **Figure R6** with the corresponding figure caption in the revised Supplementary information.

Figure R7 (Supplementary Fig. 22) XANES spectra of Au₄Pd₂/G and Au₄Pt₂/G catalysts before and after ENRR. (a) Pd K- and (b) Au L₃-edges XANES spectra of Au₄Pd₂/G catalyst, (c) Pd K-edge and (d) Au L₃-edge XANES spectra of Au₄Pd₂/G catalyst before and after ENRR, (e) Pt K-edge and (f) Au L₃-edge XANES spectra of Au₄Pt₂/G catalyst before and after ENRR.

Action: We have placed **Figure R7** with the corresponding description in the revised Supplementary information.

Supplementary Figure 17. HER performance of Au₄Pt₂/G and Au₄Pd₂/G catalysts. The onset potential of HER is determined to be ~ -0.1 V and -0.2 V for Au₄Pt₂/G and Au₄Pd₂G respectively.

Au₄Pd₂/G catalyst yields a NH₃ production rate of $13.1 \mu\text{g mg}^{-1} \text{h}^{-1}$ at -0.1V , lower than that of Au₄Pt₂/G catalyst at the same potential. This indicates that Au₄Pd₂/G has a lower ENRR activity compared to Au₄Pt₂/G. However, we obtained a maximum NH₃ yield rate of $27.1 \mu\text{g mg}^{-1} \text{h}^{-1}$ with a FE of $\sim 12\%$ at a more negative potential of -0.2 V for Au₄Pd₂/G, actually outperforming the Au₄Pt₂/G (Fig. 3) at a more negative potential.

These observations suggest that hydrogen evolution reactions (HER) could be more effectively suppressed in this system as compared to that of Au₄Pt₂/G. This is also consistent with the fact that Pt generally favors the HER, which severely limits ENRR towards NH₃ production at more negative potentials. To verify this, we have evaluated the HER performance of both Au₄Pd₂/G

and Au₄Pt₂/G catalysts (**Supplementary Fig. 17**). The results clearly demonstrate that the overpotential of Au₄Pd₂/G for HER is higher (more negative) than that of Au₄Pt₂/G, suggesting the competing HER reaction has been more effectively suppressed in the process of ENRR.

[1] We have provided the explanation for a higher NH₃ yield and faradic efficiency for Au₄Pd₂/G catalyst at more negative potential in the main text.

in page 12: “we obtained a maximum NH₃ yield rate of 27.1 μg mg⁻¹ h⁻¹ with a FE of ~12% at a more negative potential of -0.2 V for Au₄Pd₂/G, actually outperforming the Au₄Pt₂/G (Fig. 3). This suggests that HER could be more effectively suppressed in this system as compared to that of Au₄Pt₂/G, consistent with the HER performance of two bimetallic SCCs tested”

[2] A short description of HER performance of both catalysts have been included in the revised supporting information.

[3] The characterization results of Au₄Pd₂/G catalyst are included in the revised Supplementary information (Supplementary Fig. 20 c and d; Supplementary Fig. 21 a, b, c and d).

Comment 4. *The authors addressed “the 1D polymeric cluster chain was observed to disassemble into individual clusters upon dissolving in organic solvents” (Page 6). However, the experimental evidence is missing.*

Response: Thanks for the reviewer’s comment. We disagree with the reviewer regarding this point. The following experimental evidence proves the disassembly of 1D polymeric chain in the organic solvents.

[1] Firstly, the mass spectrum reveals a molecular ion peak (the signal of the largest M/Z value) at 2274 Da (Fig. 1a). This proves the dissociation of polymeric chain into individual clusters, which otherwise cannot produce such a unique feature peaked at 2274 Da

[2] One dimensional chain-like structure is expected to form on graphene if 1D polymeric chain remains intact. This is in contrast to what we have observed in TEM and AFM images (Fig 2. D and e; Supplementary Fig. 16 and Supplementary Fig. 21). These images show well-separated clusters on graphene support upon the deposition of clusters in the solution phase.

Comment 5. *On Page 3 (Supplementary Information), “The yield of Au₄Pt₂(SR)₈ is determined to be ~28%”, please describe the yield calculation method in detail.*

Response: In view of reviewer’s comment, we have provided the detailed calculation to determine the yield of Au₄Pt₂(SC₂H₄Ph)₈.

For the synthesis of $\text{Au}_4\text{Pt}_2(\text{SC}_2\text{H}_4\text{Ph})_8$ cluster, 305 mg of $\text{HAuCl}_4 \cdot 3\text{H}_2\text{O}$ (0.774 mmol Au) was used. We obtained ~126 mg of pure $\text{Au}_4\text{Pt}_2(\text{SC}_2\text{H}_4\text{Ph})_8$ (0.0554 mmol clusters, $0.0554 \text{ mmol} \times 4 = 0.22 \text{ mmol Au}$). Therefore, the yield is determined to be $0.22 \text{ mmol} / 0.774 \text{ mmol} \times 100\% = 28.4\%$ based on the mass ratio of Au species.

Comment 6. *On Page 3 (Supplementary Information), the data of single-crystal X-ray diffraction was collected at 263 K (for $\text{Au}_4\text{Pt}_2(\text{SR})_8$) and 100 K (for $\text{Au}_4\text{Pd}_2(\text{SR})_8$), respectively. Why collect single-crystal X-ray diffraction data at different temperatures?*

Response: Thank the reviewer for pointing this out.

The measurement temperature was not intentionally changed for two different crystals. In light of your comments, we have carried out SXRD measurement of $\text{Au}_4\text{Pt}_2(\text{SR})_8$ at 100 K. The corresponding two sets of data for both crystals measured at the same temperature are included in the revised Supplementary information.

Comment 7. *On Page 9 (Supplementary Information), the thermogravimetric analysis result of $\text{Au}_4\text{Pt}_2(\text{SR})_8$ crystals. The curve showed two thermal decomposition processes, and the related reaction process should be answered.*

Response: This is a good point. We would like to provide the following discussion for the explanation of our TGA result.

The first weight loss of (~37%) can be attributed to the removal of eight $-\text{C}_2\text{H}_4\text{Ph}$ organic part from thiol ligand ($-\text{S}-\text{C}_2\text{H}_4\text{Ph}$), and the subsequent weight loss (~11%) stems from the desorption of eight sulphur atoms of metal cluster. A complete loss of eight $-\text{SC}_2\text{H}_4\text{Ph}$ during TGA measurement results in a total weight loss of 48% in the whole process (Supplementary Fig. 2).

Figure R8 (Supplementary Fig. 2). Two-step weight loss with the corresponding percentage.

Comment 8. In figure 2(f), it is difficult to distinguish the arrangements of Au or Pt atoms (the plotting scale notes are missing), some high-resolution pictures should be provided.

Response: We thank the referee for this comment.

All the scale bars in Figure 2f are 5 Å. It has been included in the figure caption of 2f in the revised manuscript.

Figure 2f presents the high-resolution STEM images of anchored Au₄Pt₂ clusters on graphene. The results reveal that majority of bright dots contain a cluster of six atoms, as expected for the bimetallic Au₄Pt₂ cluster. The atomic arrangements of the imaged clusters vary, which can be attributed to the different viewing direction of the clusters or electron-beam induced cluster dissociation. Due to the close atomic mass between Pt and Au, it remains a great challenge to distinguish these two atoms based on the image contrast of STEM.

Comment 9. In Figure 6 (Supplementary Information), what is the concentration ratio of ¹⁴NH₄⁺ and ¹⁵NH₄⁺ in NMR, whether the value is proportional to the input amount of ¹⁴N₂ and ¹⁵N₂ molecules.

Response: This is a good point.

The integrated peak area associated with ¹⁵NH₄⁺ and ¹⁴NH₄⁺ is determined to be 1:9, which is proportional to the input ratio between ¹⁵N₂ and ¹⁴N₂ molecules.

Comment 10. The ammonia yields and Faradic efficiencies of multiple cycling tests should be tested.

Response: This is a good suggestion. In view of referee's suggestion, we carried out the multiple cycling tests of both Au₄Pt₂/G and Au₄Pd₂/G catalysts for ENRR.

As shown in **Figure R2**, the results reveal that both ammonia yield and FE remain nearly constant during the multiple cycling stability tests. Both catalysts demonstrate excellent stability during the process of electrochemical N₂ reduction reaction, see **Figure R2** for details.

Figure R2 (Supplementary Fig. 20). Long-term stability of both $\text{Au}_4\text{Pd}_2/\text{G}$ and $\text{Au}_4\text{Pt}_2/\text{G}$ catalysts for NRR.

Action: We have included the stability test results in the revised Supplementary information.

Comment 11. *The structural characterization of bimetallic cluster after electrochemical N₂ reduction reaction should be provided.*

Response: We thank reviewer for this valuable comment. We have conducted additional experiment to characterize the structure of the Au₄Pt₂/G and Au₄Pd₂/G after electrochemical reaction.

Figure R6 presents TEM images of both Au₄Pt₂/G and Au₄Pd₂/G catalysts before and after NRR reactions. It is observed that the cluster on graphene shows negligible aggregation and thus the size of these clusters remains nearly constant before and after reaction.

Spherical aberration corrected STEM images of both catalysts show that the clusters still consist of six-atom after NRR reaction, thus the cycling stability of both catalysts was further confirmed, (see **Figure R6**).

Additionally, XAFS measurement on Au₄Pt₂/G and Au₄Pd₂/G catalysts after ENRR was also performed. As shown in **Figure R7**, the Au, Pt L₃- edges and Pd K-edge show negligible changes after ENRR. This provides a clear evidence to support that the catalyst remains stable during the reaction.

Figure R6 (Supplementary Fig. 21). Structural characterization of bimetallic cluster after electrochemical N₂ reduction. a, b TEM images of Au₄Pt₂/G before and after ENRR. c, d TEM images of Au₄Pd₂/G before and after ENRR. (Inset b and d are spherical aberration corrected STEM images of Au₄Pt₂/G and Au₄Pd₂/G catalysts after ENRR. Scale bars for inset are 5Å)

Figure R7 (Supplementary Fig. 22) XANES spectra of Au₄Pd₂/G and Au₄Pt₂/G catalysts before and after ENRR. (a) Pd K- and (b) Au L₃-edges XANES spectra of Au₄Pd₂/G catalyst, (c) Pd K-edge and (d) Au L₃-edge XANES spectra of Au₄Pd₂/G catalyst before and after ENRR, (e) Pt K-edge and (f) Au L₃-edge XANES spectra of Au₄Pt₂/G catalyst before and after ENRR.

Comment 12. In figure 4(c)~(f), the derivative structure of partial ligand removed $Au_4Pt_2(SR)_8/G$ is incomplete at this stage, how about the comparison of experimental XANES spectra with the simulated spectra of $Au_4Pt_2(SR)_4/G$, $Au_4Pt_2(SR)_2/G$ and Au_4Pt_2/G .

Response: We appreciate the reviewer for the constructive comment.

In view of referee's comments, we have carried out modelling of additional structures ($Au_4Pt_2(SR)_4/G$, $Au_4Pt_2(SR)_2/G$ and Au_4Pt_2/G) suggested by referee. The corresponding simulated XANES spectra of these structures are shown in **Figure R9** below. Amongst all the structures proposed, the simulated spectrum of $Au_4Pt_2(SR)_6/G$ still show the best match with experimental data. Hence, our conclusion on the proposed structure ($Au_4Pt_2(SR)_6/G$) is still valid.

Figure R9 (Supplementary Fig. 10). A comparison of experimental XANES spectra with the simulated spectra of Au₄Pt₂(SR)₄/G, Au₄Pt₂(SR)₂/G and Au₄Pt₂/G.

Action: the simulated XANES spectra of Au₄Pt₂(SR)₄/G, Au₄Pt₂(SR)₂/G and Au₄Pt₂/G together with the experimental result are included in the revised supplementary information.

Comment 13. *The solvation effect could affect the barrier or free energy due to the H-bond between O atom of H₂O solvation and H atom of NRR intermediates. Thus, the authors should compare this effect on the calculation results.*

Response: Generally, it is very difficult to model solvation effect on the reaction barriers by *ab initio* calculations due to the complexity of solve involved systems. In light of your comment, we used a simple method to estimate the order of the solvation effects on the energies of the rate-limiting step, i.e. the NH₃ desorption (as shown in Fig. R10 below in the response to Comment 14 or Fig. 5d in the revised manuscript): We calculated the energy barriers of this rate-determining step with/without one water molecule in the vicinity of NH₃ molecule. Our calculations show that the barrier increases slightly by 0.06 eV with the water molecule. This result suggests that the solvation effect in the system is not significant and the major conclusion of the paper is still valid.

Comment 14. *In this study, the N₂ adsorption step is favored over the catalysts, while the desorption process of *N is not provided, and subsequent steps should be added in Figure 5d.*

Response: We appreciate the referee's constructive comments. We agree with referee that a full reaction pathway calculation is needed to provide a comprehensive understanding of the reaction mechanism. We thus carried out additional calculation of the whole reaction pathway of N₂ transformation. Our results (**Figure R10 (Fig. 5d)**) show the full reaction path for the pathway I (in which the protonation of N bonded graphene occurs first) is a more favourable route. Note that with more accurate calculations, we are able to locate a new stable state for the 4th step (2nd H in the figure) which is energetically more stable than the one showed in the previous submission. We can see from the figure that the rate-limiting step is indeed the desorption of NH₃ (the final step) with a maximum barrier of ~0.91 eV. In **Figure R11 (Supplementary Fig. 14)**, we present the full reaction path of the other mechanism in which the N atom binding with the metal cluster leaves first (the path II in the previous submission). For this mechanism, there are two rate-limiting steps: (i) the desorption of the first NH₃ with a barrier of 0.74 eV; (ii) the

formation of the second NH₃ with a high barrier of 2.42 eV. Therefore, it is most likely the reaction proceeds along the pathway I based on these calculation results.

Figure R10 (Fig. 5d). Calculated energy profile of the reaction pathway for the pathway I in which the protonation of N atom bonded graphene occurs first. Note that NH₃ form when the 3rd and 6th H added.

Figure R11 (Supplementary Fig. 14) Calculated energy profile of the reaction pathway for the alternative mechanism (pathway II in the previous submission). For this mechanism, there are two rate-limiting steps: (i) the desorption of the first NH_3 with a barrier of 0.74 eV; (ii) the formation of the second NH_3 (after the 6th H added) with a high barrier of 2.42 eV.

Comment 15. The PDOS of different orbitals for N_2 adsorbed over the catalyst has been discussed in the manuscript. As an important step, the PDOS results of catalysts with $^*\text{NNH}$ should also be added in the revised manuscript.

Response: In light of referee's comment, we calculated the PDOS of $^*\text{NNH}$ generated in the pathway I. Figure R12 reveals a significant hybridizations between N and both C and H that are bonded with N_2 , indicating that N_2 forms chemical bonds with both C and H. The formation of these bonds is energetically favourable according to the energy profile shown in Fig. R10, which is due to the fact that the N_2 molecule is already activated when adsorbed on the catalyst.

Figure R12 (Supplementary Fig. 15). Calculated PDOS of $^*\text{NNH}$ produced for the pathway under distal mechanism: (a) PDOS for all chemical species and (b) Enlarged PDOS for N, C and H that are bonded with N_2 . For a better visualization, the PDOS of H has been enhanced by 10 times.

Action: Figure R12 with the corresponding description has been included in the revised supporting information.

Reviewers' comments:

Reviewer #3 (Remarks to the Author):

The re-submitted work by Yao et al. reported electrocatalytic nitrogen reduction under ambient conditions over obtained Au₄Pt₂ and Au₄Pd₂ clusters on graphene. Although the revised draft showed an improvement over the previous one, some indispensable experiments should be added, because the authenticity of electrocatalytic ammonia synthesis has been questioned by many scholars (Nature, DOI: 10.1038/s41586-019-1260-x). Therefore, this manuscript is not recommended for publication in the present form. If the following questions can be answered satisfactorily, the manuscript will be reconsidered.

1. Although the authors conducted a mixture of ¹⁴N₂ and ¹⁵N₂ labeling to confirm the source of nitrogen for NH₃ production in Supplementary Fig. 7, it's not precise enough. Aiming toward higher accuracy and reproducibility of ENRR results, a thorough discussion of experimental parameters and testing methods in ENRR should be added, such as the reported protocols in Nature, DOI: 10.1038/s41586-019-1260-x. Chem. Soc. Rev., DOI: 10.1039/c9cs00280d. ACS Catal., 2018, 8, 7820-7827. Nature Catalysis, 2019, 2, 290. Additionally, a rigorous test procedure must be put forward that, by making essential use of ¹⁵N₂, allows the public to reliably detect and quantify the electroreduction of N₂ to NH₃.
2. It has been reported that no absorption band associated with N was observed on Pt surfaces in liquid via surface-enhanced infrared absorption spectroscopy (J. Am. Chem. Soc., 2018, 140, 1496-1501). But in this manuscript, the authors reported Pt cluster showed the ability of NRR in Supplementary Fig. 19(b). The reviewer expect the authors to use surface-enhanced infrared absorption spectroscopy to study the reaction mechanisms of NRR over obtained Au₄Pt₂ and Pt clusters.
3. The maximum nitrogen production rate is up to 23.6 μg mg⁻¹ h⁻¹ over Au₄Pt₂/G in this manuscript. The corresponding current density is not reported, nor the rate normalized to either geometric or electrochemical surface area. The actual catalyst loading mass does not have provided, so these important metrics cannot even be computed from the current data provided.
4. It is important to evaluate how much N₂ was reduced to NH₃ during the electroreduction process (so the amount of N₂ conversion rate can be evaluated). Please describe the testing process in detail.
5. Some spelling and expression errors need to be corrected, such as, "In contrast, the Au₄Pt₂/G SCC generates a maximum NH₃ yield of up to 23.6 μg mg⁻¹ h⁻¹ at 0.1 V (should be -0.1V)", "Our findings have carved out...".

Reviewer #4 (Remarks to the Author):

Single-cluster catalysis (SCC) is an emerging and exciting topic in heterogeneous catalysis, but it remains a grand challenge to synthesize SCC with atomic precise. In this respect, this work represents a breakthrough. Moreover, the Au₄Pt₂ and Au₄Pd₂ SCC show promising performances in the very demanding reaction of electrochemical nitrogen reduction. I believe this work is of highly novelty and importance, and therefore can be acceptable for publication in Nature Communication after addressing

the following concerns:

1. EXAFS fitting results need to be given to see if the Au-S and Pt-S distances as well as coordination numbers are consistent with the other characterization and theoretical results.
2. A blank test with defective graphene, instead of non-defective graphene, needs to be performed to see if this support itself is active for NRR.
3. The authors claim that the interfacial space between the graphene substrate and the Au₄Pt₂ cluster acts as the active site. What does the interfacial space? And what is difference between the interfacial space and interface?
4. From the XANES spectra, both Pt and Au atoms in the cluster catalyst are positively charged, in this case, it might not be easier to transfer electrons to N₂ molecules in comparison with negatively or zero charged metallic clusters. Please comment on this point.

Point-by-point response letter

We thank all the reviewers for their very helpful comments (*words in italics*). Below our point-by-point response is in blue.

Reviewer #3 (Remarks to the Author):

The re-submitted work by Yao et al. reported electrocatalytic nitrogen reduction under ambient conditions over obtained Au₄Pt₂ and Au₄Pd₂ clusters on graphene. Although the revised draft showed an improvement over the previous one, some indispensable experiments should be added, because the authenticity of electrocatalytic ammonia synthesis has been questioned by many scholars (Nature, DOI: 10.1038/s41586-019-1260-x). Therefore, this manuscript is not recommended for publication in the present form. If the following questions can be answered satisfactorily, the manuscript will be reconsidered.

Response: We thank the referee for your positive comments on the revised manuscript.

1. Although the authors conducted a mixture of ¹⁴N₂ and ¹⁵N₂ labeling to confirm the source of nitrogen for NH₃ production in Supplementary Fig. 7, it's not precise enough. Aiming toward higher accuracy and reproducibility of ENRR results, a thorough discussion of experimental parameters and testing methods in ENRR should be added, such as the reported protocols in Nature, DOI: 10.1038/s41586-019-1260-x. Chem. Soc. Rev., DOI: 10.1039/c9cs00280d. ACS Catal., 2018, 8, 7820-7827. Nature Catalysis, 2019, 2, 290. Additionally, a rigorous test procedure must be put forward that, by making essential use of ¹⁵N₂, allows the public to reliably detect and quantify the electroreduction of N₂ to NH₃.

Response: We thank the referee for valuable comments. We noted that several groups have published their recent work on how to design and evaluate ENRR with protocols for the precise determination of ammonia yield. These protocols allow to verify the source of ammonia. We certainly agree with the referee it can exclude the possible background/extraneous ammonia sources for the precise determination of ammonia production if we apply the same protocol published. Unfortunately, the extremely high cost of ¹⁵N₂ gas is not affordable for us to adopt the exact protocol published. In view of referee's comments, we developed a cost-effective alternative approach to further rule out the possible background/contamination ammonia sources in our experiment. The additional measurements/results prove that the data is valid and the ammonia is produced by ENRR process. We will describe our new approach and present additional data as follows.

[1] A direct comparison between the ratio of ¹⁴N₂/¹⁵N₂ gas used and the ratio of ¹⁴NH₄⁺/¹⁵NH₄⁺ produced.

The key of our method lies in the variation of the ratio of starting $^{14}\text{N}_2/^{15}\text{N}_2$ gas and check the corresponding $^{14}\text{NH}_4^+/^{15}\text{NH}_4^+$ generated after ENRR. If the correlation between these two cases can be established, this would provide a compelling evidence to support the validity of the yield and FE of ammonia generated by ENRR.

In our initial manuscript, we have conducted the $^{15}\text{N}_2$ labeling experiment to confirm the source of nitrogen for NH_3 production. Instead of the use of pure $^{15}\text{N}_2$ gas (due to the high cost), we mixed $^{15}\text{N}_2$ and $^{14}\text{N}_2$ with a mole ratio of 1:9 ($^{15}\text{N}_2/^{14}\text{N}_2$) to confirm the NH_3 production. The ratio of $^{15}\text{NH}_4^+$ and $^{14}\text{NH}_4^+$ is also determined to be 1:9 (obtained from NMR signals associated with the $^{15}\text{NH}_4^+$ and $^{14}\text{NH}_4^+$), proportional to the input gas ratio of $^{15}\text{N}_2/^{14}\text{N}_2$ (see **Figure R1** and Supplementary Fig. 7 of our original Supplementary Information). This confirms that NH_3 is indeed generated by ENRR process.

To further validate this approach, we performed additional experiment by varying the ratio of $^{15}\text{N}_2/^{14}\text{N}_2$. In this round of experiment, a ratio of 1:1 ($^{15}\text{N}_2/^{14}\text{N}_2$) was utilized as the feed gas for electrochemical reduction. The ratio of $^{15}\text{NH}_4^+/^{14}\text{NH}_4^+$ produced was determined approximately to be 1:1 using $^1\text{H-NMR}$ (**Figure R2**). All these results point out that the ammonia obtained in our experiment is derived from dinitrogen reduction rather than from the nitrogen impurities or environmental contaminations. The experimental parameters and testing methods for ENRR as well as the details of $^{15}\text{N}_2$ labeling test were included in our revised manuscript (see Method section 7-13 of the Supplementary Information).

Actions:

- In light of referee's comments, we included the detailed information regarding experimental parameters and testing methods for ENRR in the revised Supporting Information (highlighted with yellow background).
- We also include the additional data on $^{15}\text{N}_2$ labelling (Figure R2) in the manuscript and merge it into Fig 3 of the main text as panel d.

Figure R1: Supplementary Fig. 7 ^1H -NMR spectra of $^{14}\text{NH}_4^+$ and $^{15}\text{NH}_4^+$ produced from ENRR using (a) $^{14}\text{N}_2$, and (b) a mixture of $^{14}\text{N}_2$ and $^{15}\text{N}_2$ ($^{15}\text{N}_2$ enrichment ~10%). The integrated peak area associated with $^{15}\text{NH}_4^+$ and $^{14}\text{NH}_4^+$ is determined to be 1:9, which is proportional to the input ratio between $^{15}\text{N}_2$ and $^{14}\text{N}_2$ gas.

Figure R2: ^1H -NMR spectrum of $^{14}\text{NH}_4^+$ and $^{15}\text{NH}_4^+$ produced from ENRR using a mixture of $^{14}\text{N}_2$ and $^{15}\text{N}_2$ ($^{15}\text{N}_2$ enrichment ~50%). The integrated peak area associated with $^{14}\text{NH}_4^+$ and $^{15}\text{NH}_4^+$ is determined to be 1:1, which is proportional to the input ratio between $^{14}\text{N}_2$ and $^{15}\text{N}_2$ gas.

Revised Fig. 3. Electrochemical N₂ reduction reaction using different catalysts including Au₄Pt₂, Au₄Pt₂/G and Au₄Pd₂/G. (a) Schematic illustration of the experimental setup for ENRR. (b) NH₃ production rate. (c) Faradic efficiency of ammonia production at different potentials. (d) ¹H-NMR spectra of ¹⁴NH₄⁺ and ¹⁵NH₄⁺ produced from ENRR using ¹⁴N₂, mixtures of ¹⁴N₂ and ¹⁵N₂ with different ¹⁴N₂ to ¹⁵N₂ ratios. The integrated peak areas associated with ¹⁴NH₄⁺ and ¹⁵NH₄⁺ are determined to be 9:1 and 1:1 which are proportional to the input ratio between ¹⁴N₂ and ¹⁵N₂ gas.

2. It has been reported that no absorption band associated with N was observed on Pt surfaces in liquid via surface-enhanced infrared absorption spectroscopy (*J. Am. Chem. Soc.*, 2018, 140, 1496-1501). But in this manuscript, the authors reported Pt cluster showed the ability of NRR in Supplementary Fig. 19(b). The reviewer expect the authors to use surface-enhanced infrared absorption spectroscopy to study the reaction mechanisms of NRR over obtained Au₄Pt₂ and Pt clusters.

Response: We appreciate the reviewer for this good suggestion. This is a highly relevant reference (*J. Am. Chem. Soc.*, 2018, 140, 1496-1501). The authors demonstrate a nice mechanistic study of ENRR using surface-enhanced infrared absorption spectroscopy. However, Pt in the Au₄Pt₂ cluster (ionic nature, positive charge) is distinct from metallic Pt foil used in the reference. Therefore, they show different electrocatalytic activities.

In line of reviewer's suggestions, we also attempted to use the surface-enhanced infrared absorption spectroscopy (SEIRAS) to study the mechanism of the reaction. The results obtained is shown in (Figure R3). The absorption signals related to H–N–H bending, –NH₂ wagging, of adsorbed N₂H_y species are expected to occur around 1450, 1298, 1109 cm⁻¹ respectively. In this work (*J. Am. Chem. Soc.*, 2018, 140, 1496-1501), the authors studied the N absorption on a large-area Au (or Pt) thin foil (1.76 cm²) with a much higher mass of Au (or Pt) compared to that of Au (or Pt) used in our system. This leads to a much prominent signal of N-contained species. However, the mass of Au (or Pt) of the Au₄Pt₂/G (Pt/G) catalyst used for ENRR is in the microgram scale (noted 1 mg of Au₄Pt₂/G with the Au₄Pt₂(C₂H₄Ph)₈ loading of 8.5 wt% was used in the experiment). If we exclude the mass from the ligand (- C₂H₄Ph) of the cluster, the actual loading of Au and Pt is lower than 3 wt% and 1.5 wt%, respectively). Because of this, in our experiment, the signal between ~1300 and 1500 cm⁻¹ is rather featureless. It thus remains very difficult to compare the distinct IR features during ENRR. Nevertheless, we would like to mention that major focus of our work lies in the design and synthesis of atomically-precise ultrafine bimetal SCC for ENRR. A particularly interesting aspect of our findings is that the heteroatom dopant is revealed to play an indispensable role in the ENRR. The ability to precisely dope SCC raises the prospect of designing a wide range of atomically precise SCCs with dopant-controlled reactivity for broad applications beyond NH₃ production.

Figure R3. FTIR spectra during the first segment from 0.15 to -0.6 V (vs. RHE) on the Au₄Pt₂/G electrode in a N₂-saturated 0.1 M HCl solution.

Actions: We have cited the reference suggested by referee in the revised introduction part of our manuscript

3. The maximum nitrogen production rate is up to 23.6 $\mu\text{g mg}^{-1} \text{h}^{-1}$ over Au₄Pt₂/G in this manuscript. The corresponding current density is not reported, nor the rate normalized to either geometric or electrochemical surface area. The actual catalyst loading mass does not have provided, so these important metrics cannot even be computed from the current data provided.

Response: We thank the reviewer for these valuable comments. Sorry for the confusion. Some of these details were provided in the original supporting information. In view of the reviewer's comments, we have provided the yield rate (normalized by geometric area) of ammonia over Au₄Pt₂/G catalyst and loading of the metal clusters on graphene in the revised supporting information.

[1] The current (by integrating Chronoamperometric curves of an electrochemical reaction) and the area of the carbon paper ($2 \times 2 \text{ cm}^2$) were included in our supporting information (Supplementary Fig. 6a, 6c and Part 8 of Method). Therefore, the current density can be readily calculated based on these details provided. We presented current rather than current density in the report because for the calculation of Faradaic efficiency (FE) of an electrochemical reaction, the electrode area is not needed according to the formula below

$$\text{FE} = 3F \times n_{\text{NH}_3} / Q,$$

where F is the Faraday constant (96485 C mol^{-1}) and n_{NH_3} represents the mole of ammonia; Q is the total charge passed through the electrode [Current (A) \times Time (S)].

[2] In some of the previous reports, the authors reported the value normalized by the geometric area, particularly for some non-noble metal catalysts. However, we noted that the mass normalization is also widely used in some reports, in particular, for the evaluation of the electrocatalytic performance of noble metal based nanocatalysts (e.g. Nat. Commun. | 2018, 9:1795). In our work, the mass normalized yield rate was adopted to compare the catalytic performances between Au₄Pd₂/G and Au₄Pt₂/G catalysts, see **Table R1**.

[3] In line of referee's suggestions, we also determined the yield rate of ammonia normalized by geometric area (e.g. Au₄Pt₂/G as catalyst). The corresponding results were also shown in **Table R2** and **Figure R4**. We also provided the loading of metal clusters on graphene is 8.5 wt% for Au₄Pt₂/G and 10.5 wt% for Au₄Pd₂/G catalyst in the revised supporting information.

Action:

- We have included **Table R1** and **Table R2** in the revised Supplementary information (see Supplementary Table 3 and Supplementary Table 4 for details)
- The loadings of metal clusters were supplemented in the revised Supporting

information (part 8, the preparation of cathode for ENRR).

8. The preparation of cathode for ENRR. Typically, 1 mg catalyst (the loading of metal clusters on graphene is determined to be 8.5 wt% for Au₄Pt₂/G and 10.5 wt% for Au₄Pd₂/G, respectively) and 5 μ L of Nafion solution (5 wt%) were dispersed in the absolute ethyl alcohol (100 μ L) followed by the sonication for 30 minutes to form a homogeneous ink.

Table R1. The ammonia yield rate normalized by the mass (Au₄Pt₂/G).

Potential	Absorption peak	C(NH ₄ ⁺) in electrolyte	Volume of electrolyte	Mass of NH ₄ ⁺	Mass of Au ₄ Pt ₂ cluster	Mass normalized yield rate
V	a.u.	μ g/ml	ml	μ g	mg	μ g.mg ⁻¹ .h ⁻¹
-0.1	0.0270	0.0669	30	2.0087	0.085	23.63
-0.2	0.0194	0.0480	30	1.4398	0.085	16.94
-0.3	0.0165	0.0408	30	1.2239	0.085	14.39
-0.4	0.0131	0.0323	30	0.9683	0.085	11.39

Table R2. The ammonia yield rate normalized by the geometric area (Au₄Pt₂/G).

Potential (V vs. RHE)	Mass of NH ₄ ⁺ (μ g)	Geometric area (cm ²)	Time (hour)	Area normalized yield rate (μ g.h ⁻¹ .cm ⁻²)
-0.1	2.00	4	1	0.50
-0.2	1.44	4	1	0.36
-0.3	1.22	4	1	0.30
-0.4	0.97	4	1	0.24

Figure R4. The ammonia yield rate normalized by the geometric area ($\text{Au}_4\text{Pt}_2/\text{G}$).

4. It is important to evaluate how much N_2 was reduced to NH_3 during the electroreduction process (so the amount of N_2 conversion rate can be evaluated). Please describe the testing process in detail.

Response: We thank reviewer for this comment. Based on the experiment we conducted, the N_2 conversion rate is quite low ($\sim 3.5 \times 10^{-5} \%$). In the majority of previous reports on ENRR (including our experiment), N_2 gas is supplied under a continuous ventilation because of the lack of a recycle setup in the experiment so that the precise N_2 conversion cannot be determined with high accuracy.

Due to an extremely low yield of NH_3 , it also remains challenging to precisely determine the N_2 conversion rate as the accuracy of reported value varies from case by case. If one wants to obtain a higher N_2 conversion rate, it is required to reduce N_2 flow and ensure the saturation of N_2 in the electrolyte (note nitrogen can be easily saturated in aqueous electrolyte).

We would like to emphasize that the major focus of the current research (including our work) lies on the design of new efficient catalysts and fundamental study of the mechanistic insights. There is still a long way ahead towards the real-life application of electrochemical N_2 reduction.

Actions:

In view of referee's comments, we have provided the testing process in details in the revised supporting information (7-13 of Method part)

5. Some spelling and expression errors need to be corrected, such as, “In contrast, the Au₄Pt₂/G SCC generates a maximum NH₃ yield of up to 23.6 μg mg⁻¹ h⁻¹ at 0.1 V (should be -0.1V)”, “Our findings have carved out...”.

Response: We thank the reviewer for pointing out. We have revised these errors/typo in the revised manuscript.

...This work provides a new route for the design of novel SCCs with atomic precision for broad applications beyond NH₃ production.

In contrast, the Au₄Pt₂/G SCC generates a maximum NH₃ yield of up to 23.6 μg mg⁻¹ h⁻¹ at - 0.1 V...

Reviewer #4 (Remarks to the Author):

Single-cluster catalysis (SCC) is an emerging and exciting topic in heterogeneous catalysis, but it remains a grand challenge to synthesize SCC with atomic precise. In this respect, this work represents a breakthrough. Moreover, the Au₄Pt₂ and Au₄Pd₂ SCC show promising performances in the very demanding reaction of electrochemical nitrogen reduction. I believe this work is of highly novelty and importance, and therefore can be acceptable for publication in Nature Communication after addressing the following concerns:

Response: We thank the reviewer very much for the positive comments.

1. EXAFS fitting results need to be given to see if the Au-S and Pt-S distances as well as coordination numbers are consistent with the other characterization and theoretical results.

Response: thanks for this constructive suggestion. We have conducted EXAFS fitting and compared the fitting results with theoretical ones. The EXAFS fitting results show that the average bond length of both Pt-S (2.34 Å) and Au-S (2.30 Å) for Au₄Pt₂/G is longer than that of Au₄Pt₂ without graphene support, in line with the DFT calculation results (2.37 Å and 2.33 Å, respectively). Also, the coordination number of Au and Pt obtained from EXAFS fitting is in good agreement with that of DFT simulation.

Sample	Path	N	σ^2	R/ Å	R-factor
Au ₄ Pt ₂ Single crystal	Pt-S	4		2.33(0.01)	
	Au-S	2		2.31(0.01)	
Au ₄ Pt ₂ EXAFS Fitting	Pt-S	4	0.0014(0.0007)	2.31(0.01)	0.005
	Au-S	2	0.0019(0.0005)	2.29(0.01)	0.011
Au ₄ Pt ₂ /G EXAFS Fitting	Pt-S	4.5(0.9)	0.0029(0.0010)	2.34(0.02)	0.013
	Au-S	2.0(0.5)	0.0024(0.0008)	2.30(0.01)	0.006
Au ₄ Pt ₂ /G DFT	Pt-S	4		2.37	
	Au-S	2		2.33	

2. A blank test with defective graphene, instead of non-defective graphene, needs to be performed to see if this support itself is active for NRR.

Response: Thanks reviewer for pointing this out. In fact, we have conducted the control experiment using the blank defective graphene as the catalyst for ENRR. The results presented in the original manuscript, show that defective graphene alone lacks the catalytic activity for ENRR (Supplementary Fig. 6f), as supported by the UV-vis

adsorption measurement. There is no adsorption at 656 nm (indophenol method) for the electrolyte after performing ENRR for 1 hour at different applied potentials using defective graphene as catalyst, which suggests the absence of NH_3 during ENRR.

Action: To make it clearer, we revised the caption of the Supplementary Fig 6f.

Supplementary Fig. 6.....(f) Control experiments conducted using defective graphene as the catalyst.

3. *The authors claim that the interfacial space between the graphene substrate and the Au₄Pt₂ cluster acts as the active site. What does the interfacial space? And what is difference between the interfacial space and interface?*

Response: This is a good point.

Theoretical calculations reveal that the interaction between the bottom Au and Pt atoms, and carbon atoms of graphene substrate plays a crucial role in the activation and reduction of N_2 . We note that the upper Au and Pt atoms far from graphene substrate do not show any activity in the reaction. Therefore, the activation and transformation of N_2 towards NH_3 occurs in the confined space between Au₄Pt₂ cluster and graphene. Therefore, we tend to use interfacial space, although we think both terms can be used in this case (interfacial space and interface).

4. *From the XANES spectra, both Pt and Au atoms in the cluster catalyst are positively charged, in this case, it might not be easier to transfer electrons to N_2 molecules in comparison with negatively or zero charged metallic clusters. Please comment on this point.*

Response: This is a good point. Yes, it is expected that the positively charged metal

species in the cluster will behave very differently from the negatively charged or zero-valence metal clusters. If the charge transfer is mainly governed by electrostatic interactions between metal ions and electrons, we agree with referee that it would be harder to transfer electron from positively charged cluster to N_2 as compared to negatively or zero charged metallic clusters. However, in the case studied here, the clusters are supported on graphene. The charge transfer between metal atoms and the N_2 molecule is mainly controlled by the hybridization between metal and N_2 orbitals, for which the orbital alignment (rather than the electrostatic interaction) is a key factor. Our calculations (Fig. 5b and Fig. S13) have revealed that the Fermi level of graphene-supported metal clusters is very close to the LUMO of N_2 molecule so that the hybridization between N_2 LUMO and occupied metal states become possible, leading to the charge transfer.

Reviewers' comments:

Reviewer #3 (Remarks to the Author):

This is a follow up on my first review (I was reviewer #3). I have read the point-by-point response letter to both reviewers' comments carefully. The authors have tried to address all raised questions by both reviewers and to improve the presentation of their work. I also appreciate the addition of the part about the Table R1, R2 and $^{14}\text{N}_2/^{15}\text{N}_2$ labelling tests in the Supporting Information. In order to better explore the catalytic NRR reaction of $\text{Au}_4\text{Pt}_2(\text{SR})_6/\text{G}$, the authors need to answer the following questions. If these questions are answered satisfactorily, I will recommend this work to be accepted in Nat. Comm..

1. Although the authors answer the questions about NRR tests, while some tests, such as, FTIR and N_2 conversion rate did not get good results. In order to better analyze the problems from the test results and inspire other researchers in this area, the authors should analyze the reasons for the poor test results and provide improved methods. How about the interference of ligands in FTIR? What about conduct N_2 conversion rate in a closed nitrogen-saturated environment instead of flowing gas? And other possibilities? These results should be added in the revised Supporting Information.
2. Due to the severe deviation of the AFM test baseline, the author should retest the particle size in Supplementary Fig. 5.
3. The abbreviations for catalysts are confusing, the as-prepared catalyst for ENRR is " $\text{Au}_4\text{Pt}_2(\text{SR})_6/\text{G}$ " instead of $\text{Au}_4\text{Pt}_2/\text{G}$. And the similar abbreviations should be revised.

Reviewer #4 (Remarks to the Author):

The authors have provided a lot of new data for addressing the main concerns of the referees and the revised manuscript is of high quality. I am satisfied with the responses and recommend its publication without further revision.

Point-by-point response letter

We thank all the reviewers for their constructive comments (*words in italics*). We have provided our point-by-point response to each comment (in blue). The revised part for the Main Text and Supporting Information are marked with yellow background.

Reviewer #3 (Remarks to the Author):

This is a follow up on my first review (I was reviewer #3). I have read the point-by-point response letter to both reviewers' comments carefully. The authors have tried to address all raised questions by both reviewers and to improve the presentation of their work. I also appreciate the addition of the part about the Table R1, R2 and 14N2/15N2 labelling tests in the Supporting Information. In order to better explore the catalytic NRR reaction of Au4Pt2(SR)6/G, the authors need to answer the following questions. If these questions are answered satisfactorily, I will recommend this work to be accepted in Nat. Comm.

Response: We highly appreciate the reviewer for the positive comments and valuable suggestions.

1. Although the authors answer the questions about NRR tests, while some tests, such as, FTIR and N2 conversion rate did not get good results. In order to better analyze the problems from the test results and inspire other researchers in this area, the authors should analyze the reasons for the poor test results and provide improved methods. How about the interference of ligands in FTIR? What about conduct N2 conversion rate in a closed nitrogen-saturated environment instead of flowing gas? And other possibilities? These results should be added in the revised Supporting Information.

Response: We appreciate the reviewer for these valuable comments.

In line of referee's comments, we have carried out additional experiment and provided the possible reasons for FTIR results. Our new FTIR measurements of $\text{Au}_4\text{Pt}_2(\text{SR})_8$ clusters (**Figure R1**) confirm that **the presence of ligands does interfere the FTIR results** (as pointed by referee). As shown in Figure R1, some distinct absorption features dominate in the spectrum range from ~ 1250 to 1600 cm^{-1} (such as aromatic C–C stretching at around 1450 , 1500 , 1580 and 1600 cm^{-1} and $\text{CH}_2\text{--CH}_2$ bending at around 1272 , 1355 cm^{-1}). [Ref 1: *Phys. Chem. Chem. Phys.*, 2013, 15, 12539-12542. Ref 2: *Spectrochimica Acta Part A* 2011, 82, 63–68] Unfortunately, the absorption features related to adsorbed N_2H_y species are also expected to occur in the same spectrum range (**Figure R2**), which renders it very challenging to observe noticeable variation of the IR features related to N_2H_y species during ENRR.

In addition, although the total mass of Au (or Pt) of the $\text{Au}_4\text{Pt}_2(\text{SR})_6/\text{G}$ catalyst used for ENRR lies in the microgram scale, the actual loading of Au and Pt metal species is lower than 3 wt% and 1.5 wt%, respectively if the mass contributed from the cluster ligand (- $\text{C}_2\text{H}_4\text{Ph}$) is excluded. A low metal loading (a low density of active site) generally results in a weaker signal, consistent with our FTIR observation (Figure R2)

Figure R1. FTIR spectra of Au₄Pt₂(SR)₈. The signals between ~1250 and 1500 cm⁻¹, overlap with the absorption signals related to adsorbed N₂H_y species.

Figure R2. FTIR spectra during the first segment from 0.15 to -0.6 V (vs. RHE) on the Au₄Pt₂(SR)₆/G electrode in a N₂-saturated 0.1 M HCl solution.

In view of referee's comments, we also conducted the electrochemical N₂ reduction using a home-design equipment under a closed nitrogen-saturated environment (The N₂ volume is 2 L). The concentration of NH₄⁺ was determined as 5.31 μg/mL by using indophenol blue method (**Figure R3**) after the reaction has been carried out for four days. The total volume of electrolyte is 30 mL and thus the total mass of NH₄⁺ is calculated to be 159.32 μg. In this case, the N₂ conversion rate is determined to be ~ 4.96×10⁻³ % (**Table R1**).

Figure R3. UV-Vis absorption spectrum of the electrolyte after performing ENRR for four days using Au₄Pt₂/G catalyst (indophenol method. Inset: cuvettes with electrolyte with/without electrochemical reaction).

Table R1. The N₂ conversion rate in a closed nitrogen-saturated environment.

Experimental Absorbance intensity	Absorption intensity at C(NH ₄ ⁺)=0.4μg/mL	Volume of electrolyte (mL)	m(NH ₄ ⁺) (μg)	Volume of N ₂ (L)	N ₂ conversion rate (%)
2.14922	0.16188	30.0	159.31949	2.0	4.96×10 ⁻³

Action:

We have placed additional data/figures including **Figure R1**, **Figure R2**, **Figure R3**, and **Table R1** in the revised Supplementary information (see **Supplementary Extended Data Fig. 1**, **Fig. 2**, **Fig. 3** and **Table 1**).

2. Due to the severe deviation of the AFM test baseline, the author should retest the particle size in Supplementary Fig. 5.

Response: We have conducted AFM measurement to better estimate the cluster size (height) according to referee's suggestion.

AFM image in (**Figure R4**, also Supplementary Fig. 5) reveals the morphology of $\text{Au}_4\text{Pt}_2(\text{SR})_8$ clusters deposited on monolayer graphene/ SiO_2 . The surface roughness of SiO_2 substrate (0.5-1 nm) results in a slight deviation of the cluster height measured by AFM height profile. To solve this issue, we also conducted AFM measurement of $\text{Au}_4\text{Pt}_2(\text{SR})_8$ clusters deposited on HOPG. The apparent height of individual clusters is determined to be 1.8 nm (**Figure R5**), in good agreement with the theoretical value obtained from single crystal structure analysis for individual clusters (1.6 nm \times 1.6 nm \times 2.0 nm).

Figure R4 (Supplementary Fig. 5). AFM image of $\text{Au}_4\text{Pt}_2(\text{SR})_8$ clusters deposited on graphene/ SiO_2 . The inset shows an AFM height profile acquired along the dotted green line.

Figure R5. AFM image of Au₄Pt₂(SR)₈ clusters deposited on HOPG. The inset represents an AFM height profile acquired along the white line.

Action: We have replaced the original AFM data (**Figure R4**) with this one (**Figure R5**) in the revised Supplementary information (see Supplementary Fig. 5).

3. *The abbreviations for catalysts are confusing, the as-prepared catalyst for ENRR is “Au₄Pt₂(SR)₆/G” instead of Au₄Pt₂/G. And the similar abbreviations should be revised.*

Response: This is a good suggestion.

We used accurate molecular formula (Au₄Pt₂(SR)₈ or Au₄Pd₂(SR)₈) to illustrate individual cluster before its deposition on graphene substrate. When these clusters are landed on graphene, they undergo the partial removal of ligands. However, the exact number of ligands of anchored clusters on graphene remains unknown till we carried

out the DFT calculation in combination with the standard EXAFS fitting of proposed structures. Therefore, we used Au₄Pt₂/G (or Au₄Pd₂/G) to represent single cluster catalysts prepared, prior to the detailed structural analysis. After we have identified the number of ligands of anchored Au₄Pt₂/G (or Au₄Pd₂/G) on graphene, we used Au₄Pt₂(SR)₆/G (or Au₄Pd₂(SR)₆/G) for a higher accuracy.

In view of referee's comments, we have specified it and made it consistent in the revised manuscript.

Reviewer #4 (Remarks to the Author):

The authors have provided a lot of new data for addressing the main concerns of the referees and the revised manuscript is of high quality. I am satisfied with the responses and recommend its publication without further revision.

Response: We highly appreciate the reviewer for the positive comments.

REVIEWER COMMENTS

Reviewer #3 (Remarks to the Author):

There are still some problems in the revised manuscript, I would recommend this work for publication after major revision according to the following issues:

1. In this manuscript, the author should provide a visualized characterization of the defects in Au₄Pt₂/G catalysts.
2. Through STM and STS results of individual Au₄Pt₂(SR)₈ clusters deposited on a graphite surface, the authors indicated “the presence of weak interactions between the cluster and substrate”. The authors should point out the type of graphene here (defective or perfect), and provide the synthesis method of the catalyst.
3. The authors should provide the synthesis method of Au₄Pt₂/G in this manuscript.
4. In this manuscript, the LSV curves of Au₄Pt₂/G specimen in the atmosphere of nitrogen and argon should be added to determine the onset electrolytic potential of NRR.
5. In Figure 3b-c, the trend of Faraday efficiency and NH₃ yield rate of Au₄Pt₂(SR)₈ and Au₄Pt₂/G does not show the peak value from -0.1 to -0.4 V, the authors should expand the range of electrolysis to determine the optimum voltage.
6. In the manuscript, the Au/Pt atomic ratio of Au₄Pt₂/G is 4:2. Whether the authors have tried other atomic ratios? In general, the proportion of metals in the bimetallic material would affect the performance of the catalyst.
7. In Supplementary Figure 10, the authors found that “N₂ adsorption cannot proceed over any site of Au₄Pt₂(SR)₆ once it is anchored on defect-free graphene”. It is better to provide the relevant electrochemical performances of Au₄Pt₂(SR)₆ on defect-free graphene.

Point-by-point response letter

We thank the reviewer for her/his comments (words in italics). We have provided our point-by-point response to each comment (in blue). The revised part for the Main Text and Supporting Information are marked with yellow background.

Reviewer #3 (Remarks to the Author):

There are still some problems in the revised manuscript, I would recommend this work for publication after major revision according to the following issues:

1. In this manuscript, the author should provide a visualized characterization of the defects in Au₄Pt₂/G catalysts.

Response: It is extremely challenging to visualize graphene's defect presented in Au₄Pt₂/G catalyst because these defects are bonded with clusters. The contrast of STEM imaging will be dominated by the heavy metal atoms at active sites rather than the defect in graphene. STM technique cannot be used to characterize the powder sample. However, the presence of vacancy defect in graphene have been reported in numerous previous work.[*Nat. Commun.*, **2012**, *3*, 1144; *Angew. Chem. Int. Ed.*, **2019**, *58*, 1163–1176] Vacancy in graphene has also been widely used to anchor single-atoms for catalysis.[*Chem. Rev.* **2019**, *119*, *3*, 1806–1854; *Nat. Commun.* **2017**, *8*, 1070; *Nat. Commun.* **2018**, *9*, 3197] To verify the validity of our proposed structure, we used X-ray Absorption Fine Structure (XAFS) technique in combination with a theoretical simulation, a standard method to confirm the structure of active sites in this field.[*Nat. Commun.* **2018**, *9*, 3197; *Nat. Mater.* **2015**, *14*, 937– 942; *JACS*, **2020**, *142*, *12*, 5709–5721]

2. Through STM and STS results of individual Au₄Pt₂(SR)₈ clusters deposited on a graphite surface, the authors indicated “the presence of weak interactions between the cluster and substrate”. The authors should point out the type of graphene here

(defective or perfect), and provide the synthesis method of the catalyst.

Response: it is a high-quality graphite substrate, namely highly oriented pyrolytic graphite (HOPG), which provides a clean and ideal surface for STM imaging. The surface of HOPG can be viewed as a perfect graphene.

We performed scanning tunneling microscope (STM) imaging and spectroscopy (STS) measurement (by dissolving the $\text{Au}_4\text{Pt}_2(\text{SR})_8$ clusters into dichloromethane (DCM) and drop-casting the clusters on HOPG) to investigate the structure and electronic properties of individual $\text{Au}_4\text{Pt}_2(\text{SR})_8$ clusters. It is worth noting that, due to the weak interactions between the cluster and substrate, HOPG (perfect graphene without defect) cannot be used as substrate for the fabrication of SCCs.

3. The authors should provide the synthesis method of $\text{Au}_4\text{Pt}_2/\text{G}$ in this manuscript.

Response: Thanks for pointing out. We have included a more detailed synthesis method of $\text{Au}_4\text{Pt}_2/\text{G}$ in the revised Supplementary materials, see method 1-8.

4. In this manuscript, the LSV curves of $\text{Au}_4\text{Pt}_2/\text{G}$ specimen in the atmosphere of nitrogen and argon should be added to determine the onset electrolytic potential of ENRR.

Response: Thanks. We have done LSV measurement for $\text{Au}_4\text{Pt}_2/\text{G}$ catalyst before we perform ENRR test. As shown in the **Figure R1**, the onset electrolytic potential of ENRR is around -0.1 V (vs. RHE).

Figure R1. LSV curves of Au₄Pt₂/G catalyst in argon and nitrogen, respectively.

5. In Figure 3b-c, the trend of Faraday efficiency and NH₃ yield rate of Au₄Pt₂(SR)₈ and Au₄Pt₂/G does not show the peak value from -0.1 to -0.4 V, the authors should expand the range of electrolysis to determine the optimum voltage.

Response: Based on the LSV result above, the onset potential of ENRR should be around -0.1 V. Therefore, we performed chronoamperometry analysis in the potential range from -0.1 to -0.4 V. We have done the electrolysis at 0 V vs RHE, but no ammonia was detected using indophenol blue method (see **Figure R2**). Therefore, this data was not shown in our manuscript.

Figure R2. UV-Vis absorption spectrum of the electrolyte after performing ENRR for

1 hour at 0 V (vs. RHE) using Au₄Pt₂/G catalyst (indophenol method).

6. *In the manuscript, the Au/Pt atomic ratio of Au₄Pt₂/G is 4:2. Whether the authors have tried other atomic ratios? In general, the proportion of metals in the bimetallic material would affect the performance of the catalyst.*

Response: We agree with referee that the proportion of metals in bimetallic materials will affect their performance of catalysis. However, the synthesis of atomically precise metal clusters is completely different from the synthesis of normal metal nanoparticles. First, it is not feasible to tune the composition of clusters by simply adjusting the ratio of precursors. Second, it is extremely challenging (impractical) to vary the Au/Pt ratio under the same structural framework. Since the coordination chemistry of Au and Pt is very different, it is very likely that Au-Pt clusters with other atomic ratios will show a completely different structure configuration as compared to Au₄Pt₂/G.

This work focuses on a systemic study of the ENRR property of bimetallic SCCs and their catalytic mechanism as well as the hetero-atom doping effect on the catalysis (compare Au₄Pt₂/G and Au₄Pd₂/G). The study of the catalytic property of bimetallic Au-Pt cluster with different ratios deserves a future study but it is beyond the scope of this work.

7. *In Supplementary Figure 10, the authors found that “N₂ adsorption cannot proceed over any site of Au₄Pt₂(SR)₆ once it is anchored on defect-free graphene”. It is better to provide the relevant electrochemical performances of Au₄Pt₂(SR)₆ on defect-free graphene.*

Response: Both experimental and theoretical results indicate that the defective graphene plays an important role in the fabrication of Au₄Pt₂/G catalyst. Experimental and simulated XANES reveal a stable structure consisting of partially ligand-protected Au₄Pt₂(SR)₆ bonded to graphene vacancy, whereby Pt-C anchoring bond can be formed after a removal of two ligands at the base of each Au₄Pt₂(SR)₈ cluster. In our theoretical

calculation, we also found that N_2 adsorption cannot proceed over $Au_4Pt_2(SR)_6$ anchored on defect-free graphene. This is a purely theoretical model rather than a real catalyst. Such a theoretical model (*$Au_4Pt_2(SR)_6$ on defect-free graphene*) cannot be experimentally realized to conduct electrochemical test.

REVIEWER COMMENTS

Reviewer #3 (Remarks to the Author):

The authors have made appropriate supplements and reasonable explanations. Considering the good results, the revised manuscript is now acceptable.